# Templated growth of oriented layered hybrid perovskites on 3D-like perovskites

Jifei Wang [1,7], Shiqiang Luo [1,7], Yun Lin [2,7], Yifu Chen [1,7], Yehao Deng [2], Zhimin Li [3], Ke Meng [3], Gang Chen [3], Tiantian Huang[4], Si Xiao [1], Han Huang [1], Conghua Zhou[1], Liming Ding [5], Jun He[1], Jinsong Huang [2]* & Yongbo Yuan [1,6]*

The manipulation of crystal orientation from the thermodynamic equilibrium states is desired in layered hybrid perovskite films to direct charge transport and enhance the perovskite devices performance. Here we report a templated growth mechanism of layered perovskites from 3D-like perovskites which can be a general design rule to align layered perovskites along the out-of-plane direction in films made by both spin-coating and scalable blading process. The method involves suppressing the nucleation of both layered and 3D perovskites inside the perovskite solution using additional ammonium halide salts, which forces the film formation starts from solution surface. The fast drying of solvent at liquid surface leaves 3D-like perovskites which surprisingly templates the growth of layered perovskites, enabled by the periodic corner-sharing octahedra networks on the surface of 3D-like perovskites. This discovery provides deep insights into the nucleation behavior of octahedra-array-based perovskite materials, representing a general strategy to manipulate the orientation of layered perovskites.

[1] Hunan Key Laboratory of Super Microstructure and Ultrafast Process, School of Physics and Electronics, Central South University, Changsha, Hunan 410083, P. R. China. [2] Department of Applied Physical Sciences, University of North Carolina at Chapel Hill, Chapel Hill, NC, USA. [3] School of Physical Science and Technology, Shanghai Tech University, Shanghai 201210, China. [4] State Key laboratory of high performance complex manufacturing, School of Mechanical and Electrical Engineering, Central South University, Changsha, Hunan 410083, P. R. China. [5] Center for Excellence in Nanoscience (CAS), Key Laboratory of Nanosystem and Hierarchical Fabrication (CAS), National Center for Nanoscience and Technology, Beijing 100190, China. [6] State Key Laboratory of Powder Metallurgy, Central South University, Changsha, Hunan 410083, P. R. China. [7] These authors contributed equally: Jifei Wang, Shiqiang Luo, Yun Lin, Yifu Chen. *email: yuanyb@csu.edu.cn; jhuang@unc.edu

Layered hybrid perovskites such as Ruddlesden-Popper (RP) perovskites and Dion-Jacobson (DJ) perovskites have attracted tremendous attentions due to their superior moisture stabilities, thermal stabilities, and suppressed ion migration than their 3D counterparts[1–6]. However, layered perovskites are highly electrically anisotropic, because the charge transport along out of plane (OP) direction is much hindered by the low-conducting organic spacing layers[7,8]. Therefore, manipulating the orientation of layered perovskites becomes vital due to its significant impacts on the power conversion efficiency (PCE) of the resulted solar cells. Nevertheless, the orientation of layered perovskites might not be as desired, which is determined by both the thermodynamics of material structure as well as the kinetics in material formation process. Additionally, in-plane orientation (IP) orientation is common for two-dimensional (2D) materials with electron-rich structure due to the Van der Waals force or/ and electrostatic induction between 2D materials and conductive substrates[9–12]. For the butylamine (BA) based RP perovskites, the BA-terminated planes have the lowest surface energy, which lead to IP of this layered perovskite on many commonly used substrates to minimize the interface energy in the material system[7,13]. On the other hand, OP orientation was reported to form in films made by some special formation processes, such as hot casting method[1], despite that the OP orientation is rarely seen in RP perovskites with low layer number, such as $n = 1$[3,14,15]. However, up to now, knowledge about manipulating the plane orientation of layered perovskites is still sparse. The driving force for the conditional OP orientation is unknown.

Generally, entropic force[16–18] and chemical bonding effect[19] can drive the orientation of polygonal crystals or shaped particles in solutions, leading to varied assembling or orientation behaviors (Supplementary Note 1). However, the role of these driving forces in the crystallization of perovskite materials is less investigated. The nucleation and growth of perovskite crystals in liquid phase could be rather complex since it is ternary (or polynary) system. Due to this complexity, it is essential to establish a model that takes the predominant effects into account and build a general frame to understand the crystallography of octahedra-array-based perovskites.

In this contribution, we systematically investigate the formation process of layered perovskites and clarify a key driving force that dominates the nucleation and directional growth of layered perovskite crystals, which further shows its broad validity in the manipulation of the crystallinity and orientation of different types of layered perovskites fabricated by spin-coating method and/or doctor blading method.

**Formation of RP layered perovskites with OP orientation**. RP perovskite with stoichiometric ratio of $(BA)_2(MA)_{n-1}Pb_nI_{3n+1}$ (e.g., average layer number $<n> = 4$) were mainly focused on (Fig. 1 and Supplementary Fig. 1) in this study. As inspired by the success of $NH_4Cl$ additive in promoting the crystallinity of 3D perovskites[20], recently $NH_4Cl$ additive was also employed in RP perovskite for grains with OP orientation[21,22]. By pre-mixing $NH_4Cl$ in precursors (with a molar ratio of $NH_4Cl:PbI_2 = 0.5$), the formation of dense RP perovskite films from a simple spin-coating method was realized. The impact of $NH_4Cl$ on crystal orientation was investigated by grazing-incidence wide-angle X-ray scattering (GIWAXS) patterns (Fig. 1a, b). The ring-shaped diffraction pattern in Fig. 1a indicated a random crystalline orientation in RP perovskite film without $NH_4Cl$. Furthermore, the strong (0 2 0) and (0 4 0) peaks along $q_z$ axis suggested a significant IP orientation. In contrast, the highly concentrated diffraction spots for the RP perovskite film with $NH_4Cl$ additive indicated a much ordered crystal orientation. The clear exciton

absorption peaks for layered perovskite (Supplementary Fig. 1a) together with the absence of diffraction peaks along $q_z$ axis in the range of $0 \sim 10 \, nm^{-1}$ (Fig. 1b) suggests the dominating OP orientations (Fig. 1c) in RP perovskite film with $NH_4Cl$ additive[1,23]. In this study, OP orientation of RP perovskites is achieved in films on both planar PEDOT:PSS surface and mesoporous $TiO_2$ surface (Supplementary Fig. 2). It can be confirmed that the rough $TiO_2$ surface with disordered normal directions has no impact on the $NH_4Cl$ induced OP orientation.

**Mechanism for directional growth of RP layered perovskites**. The changing of dominating crystal orientation from IP to OP by $NH_4Cl$ additive offers an excellent platform for in-depth investigation of the nucleation and growth of RP perovskites[21,22]. We thus look into the underlying mechanism for the OP orientation of RP perovskites. In a previous study, the earlier formation of RP perovskite crystals at the liquid-air interface was reported to induce a downward growth of RP layered perovskites during the solution drying, which was explained as the origin of OP grain formation[24]. However, we observed that the RP crystals formed at the liquid-air interface could also be IP orientation. As shown in Fig. 2a, b for a slowly cooled oversaturated RP precursor solution ($<n> = 2$) without external disturbance, large RP perovskite single crystals grew horizontally at the liquid-air interface, which was confirmed to be IP orientation (Supplementary Fig. 3). This IP orientation is reasonable because the low surface energy of RP single crystal surface terminated with alkyl chains favors IP alignment[25]. This result suggests that the OP crystal orientation is not directly determined by the preferential formation of RP crystals at liquid-air interface[24], at which region entropic force would also cause IP orientation[15–17,26].

Since the precursor solution is a mixture from $PbI_2$, BAI, and MAI raw materials, the solubility of these three raw materials in DMF follows a trend of $BAI > MAI > PbI_2$ (Fig. 2b). A fact we noticed is that $PbI_6$ octahedral colloids prefer to precipitate firstly from the solution due to its much lower solubility in DMF (~1.0 M) by forming one-dimensional $PbI_2$-DMF-contained solvate phases (PDS)[27,28], which can be observed under optical microscope (inset of Fig. 2b). The PDS formed in MAI-contained solution can be a mixture of $PbI_2$-DMF and $(MA)_2(DMF)_2$ $Pb_mI_{2m+2}$ ($m = 2,3$) phases (see Supplementary Figure 4 and Supplementary Note 2), the latter of which has been reported to be the intermediate phase for the formation of perovskites[29]. Some in situ GIWAXS studies also proved the presence of solvate phase before perovskite formation[27,28]. Based on this hint, experiments were carried out to explore whether the preformed PDS impacts on the following nucleation and directional growth of RP perovskites during solution thinning. Since in situ observing the nucleation process in nanometer scale inside liquid phase is highly challenging, in our study, some $PbI_2$-DMF powders were intentionally dropped on top of the oversaturated RP precursor solution to create observable PDS phase (Fig. 2c, d). When soaked in oversaturated RP precursor solution, the surface of PDS powders turn black in a few seconds (Supplementary Fig. 5). The red shift of the absorption onset from ~550 to ~750 nm (Fig. 2e) and the photoluminescence (PL) peak around 750 nm (Fig. 2f) suggests the formation of 3D-like corner-sharing $PbI_6$ octahedra networks with reduced bandgap. We refer it as 3D-like perovskite, because this corner-sharing $PbI_6$ octahedra networks are less ideal than the 3D octahedra networks in tetragonal $MAPbI_3$ perovskite due to the presence of absorbed DMF molecules and less continuous in structure. These 3D-like perovskites are converted from the double-chains of edge-sharing Pb-I based octahedra in $PbI_2$-DMF[30], or the triple-chains of edge-sharing $PbI_6$ octahedra in $MA_2(DMF)_2Pb_3I_8$ as an intermediate

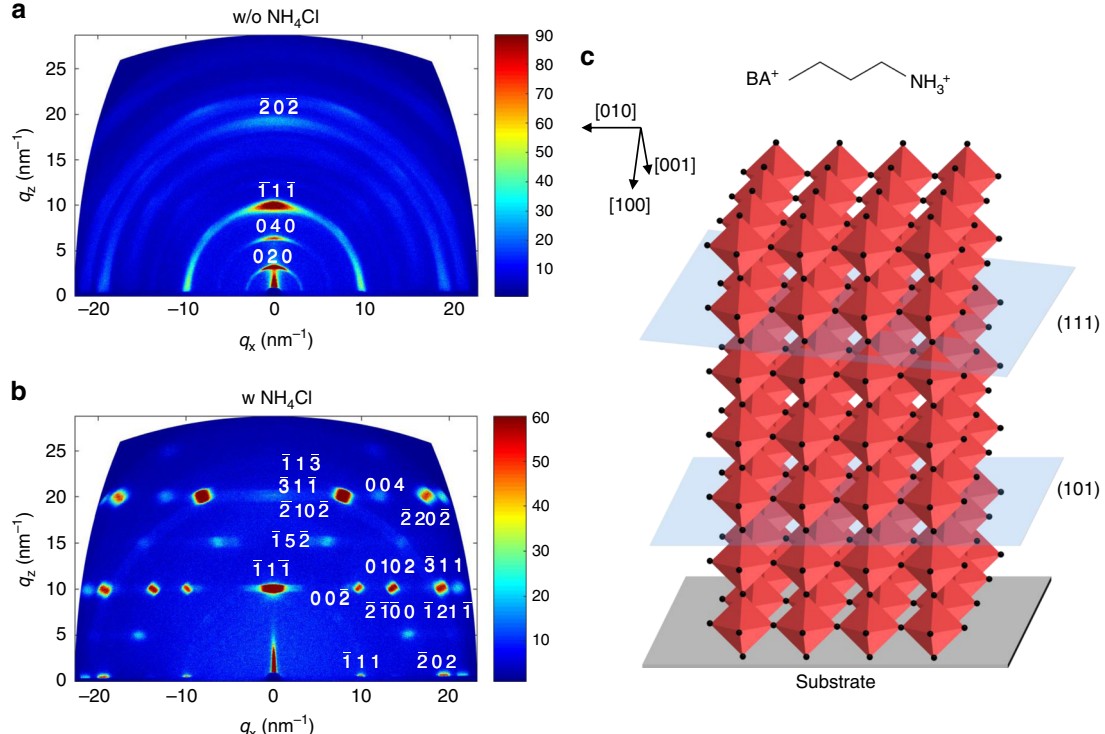

**Fig. 1 Effect of NH₄Cl additive on crystallization and orientation. a, b** GIWAXS patterns of BA-based RP films (<*n*> = 4) spun from precursor solution without additive (**a**) or with NH₄Cl additive **(**molar ratio of NH₄Cl/PbI₂ = 0.5) (**b**). **c** Illustration of BA molecular structure and RP perovskites with out-of-plane orientation.

phase[29], based on our XRD study shown in Supplementary Figure 5 and Supplementary Note 3 (the geometric relationship between the edge-sharing octahedra chains and corner-sharing octahedra chains will be discussed below). Interestingly, RP perovskite crystals were found to predominantly grow underneath the 3D-like perovskites coated PDS and adopt OP orientation (Fig. 2c, d). In another demonstration, a PDS fiber made of PbI₂-DMF nanowires was soaked into the oversaturated RP precursor solution (Fig. 2g). The surface again converted to black colored 3D-like perovskites quickly and followed by the growth of RP perovskite flakes on the surface. Most of the RP crystal flakes have an orientation perpendicular to the PDS fiber surface (Fig. 2g). This result reveals the strong correlation between the presence of 3D-like perovskites and the directional growth of the RP perovskites. As shown in Figs. 2d and g, almost all the formed RP crystals are grown from 3D-like perovskites coated PDS when the precursor concentration do not exceed the critical nucleation concentration, indicating the nucleation of RP perovskite on 3D-like perovskites is energetically favored compared to the homogenous nucleation from inside of the solution. The growth rate of the directional RP perovskites on PDS fiber is estimated to be ~1.0 μm s⁻¹ (Supplementary Fig. 6). The high speed enables the formation of RP perovskites within one second in regular perovskite films which generally have a thickness of two to three hundred nanometers. Replacing the PbI₂-DMF phase in Fig. 2d,g with (MA)₂(DMF)₂Pb₃I₈ phase or PbI₂-DMSO based solvated phase can also lead to same directional growth of RP perovskites (Supplementary Figs. 7 and 8), this is because those solvated phase has similar double/triple chains of edge-sharing PbI₆ octahedra to form 3D-like perovskite on its surface[29]. For a further demonstration of the efficient directional growth of RP perovskites from 3D-like perovskites, we intentionally accelerate the solidifying process by directly dropping oversaturated RP precursor solution into

chlorobenzene (CB) antisolvent, which result in the formation of many particles in the micrometer scale (Supplementary Fig. 9 and Supplementary Note 4). These particles were found to be 3D-like perovskites coated PDS particles with RP perovskite flakes growing from edges (Fig. 2h, i) as identified by atomic force microscopy-infrared spectroscopy (IR-AFM) and energy dispersive spectroscopy (EDS, Supplementary Fig. 9 and Supplementary Note 4). This experiment further visualized the nucleation and preferred OP growth of RP perovskites triggered by preformed 3D-like perovskites phase, which is difficult to be directly recognized in spin-coated films by cross-sectional SEM image method, because the RP perovskites nucleation process is transient, i.e., the structures related to the nucleation process will be covered by the subsequently formed crystals.

Discovering the 3D-like perovskites triggered directional crystal growth provides a framework to understand the conditional OP orientation of RP perovskites. The obtained layered perovskite films can result from several competing growth modes. The reason for the OP orientation growth become dominating in Fig. 1 is that NH₄Cl additive suppress the nucleation of PDS (and hence 3D-like perovskites) inside the solution, while only the 3D-like perovskites formed at the solution surface due to solvent evaporation can seed the growth of layered perovskites. As illustrated in Fig. 3a, in the case of RP precursor solution without NH₄Cl, too many PDS microcrystals form simultaneously at the liquid-air interface and inside solution due to its poor solubility and too much PDS materials supply during solution thinning. The PDS hence grow rapidly in oversaturated solution and stack randomly. The subsequent 3D-like perovskites formed on random-oriented PDS surface then causes the growth of RP perovskites with random orientations. It's not necessarily all the PDS will be converted into perovskite phase during solution thinning, depending on the dynamic of DMF evaporation. So that the PbI₂-DMF (and (MA)₂(DMF)₂PbₘI₂ₘ₊₂) can be detected in

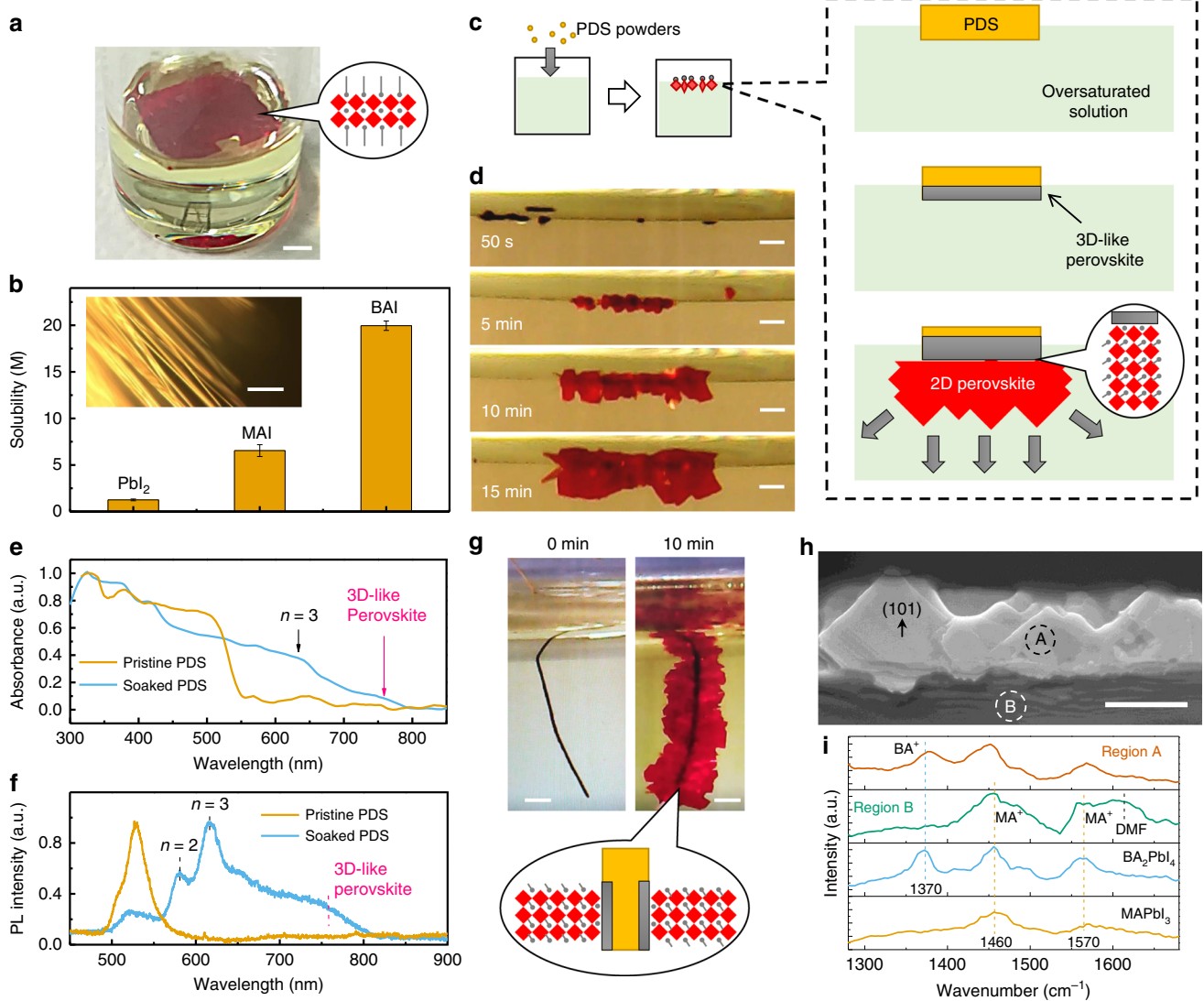

**Fig. 2 Templated growth of layered perovskites on 3D-like perovskites. a** $BA_2MAPb_2I_7$ single crystal formed on the air/liquid interface by slowly cooling supersaturated solution. The scale bar is 2 mm. **b** Comparison of the solubility of $PbI_2$, MAI, and BAI in DMF solvent, where the insets shows the first precipitation of $PbI_2$-DMF-contained solvate phases (PDS) in BA-based RP perovskite precursor solution during drying. The error bars represent the s.d. of three measurements. The inset scale bar is 10 μm; **c, d** illustration (**c**) and optical photos (**d**) of the directional growth of BA-based RP perovskite crystals under the intentionally spread $PbI_2$-DMF powder at the liquid surface of the oversaturated precursor solution ($<n> = 2$). The scale bar is 0.5 mm. The right panel illustrates the formation of 3D-like perovskites on soaked PDS surface, which triggers directional growth of RP perovskites; **e, f** optical absorption spectrum (**e**) and PL spectrum (**f**) of soaked $PbI_2$-DMF with 3D-like perovskites formed on surface; **g** optical photos of the directional growth of BA-based RP crystals on $PbI_2$-DMF fibers soaked in the oversaturated precursor solution ($<n> = 2$). The scale bar is 0.5 mm. **h** SEM image of RP perovskite grow from 3D-like perovskites coated PDS. The scale bar is 10 μm; **i** localized infrared spectra of the RP perovskites and 3D-like perovskites coated PDS (e.g., the region A and region B marked in (**h**), respectively) as measured by IR-AFM.

spin-coated film (Supplementary Fig. 10) until heated at elevated temperature of 70~100 °C[29]. As a contrast, when $NH_4Cl$ additives are introduced, the precipitation of PDS in solution is much suppressed due to the enhanced solubility of $PbI_6$ octahedral colloids by $NH_4Cl$ (Fig. 3b), which has been proved experimentally in our study (Fig. 3c and Supplementary Note 5). The much reduced homogenous nucleation of PDS inside solution makes the precipitation of PDS at the top of the liquid phase dominating. This is because the evaporation of DMF near the liquid surface is rapid during high-speed spin coating. The preformed PDS microcrystals then trigger overwhelmingly downward growth of RP perovskites. The proposed growing model and corresponding resulted morphologies in Fig. 3a, b are consistent with the measured cross-sectional SEM images of the

samples without (Fig. 3d) and with $NH_4Cl$ additives (Fig. 3e), respectively. Since the preformed PDS are much suppressed by $NH_4Cl$ additives, these residual solvated phases may not be necessarily detectable by XRD (Supplementary Fig. 10).

More generally, due to the competition between the downward growth of RP perovskites from 3D-like perovskites on top and the random growth of RP perovskites from 3D-like perovskites in the bulk of liquid phase, engineering the location and amount of preformed PDS and 3D-like perovskites is a straightforward method to promote OP orientation of RP perovskites, i.e., those methods help to reduce the nucleation of PDS inside the solution, such as using hot solution[1,2,23,31], adding DMSO[31,32], or using other organic solvents[24] would promote OP orientation. For a further demonstration, our additional GIWAXS and XRD studies

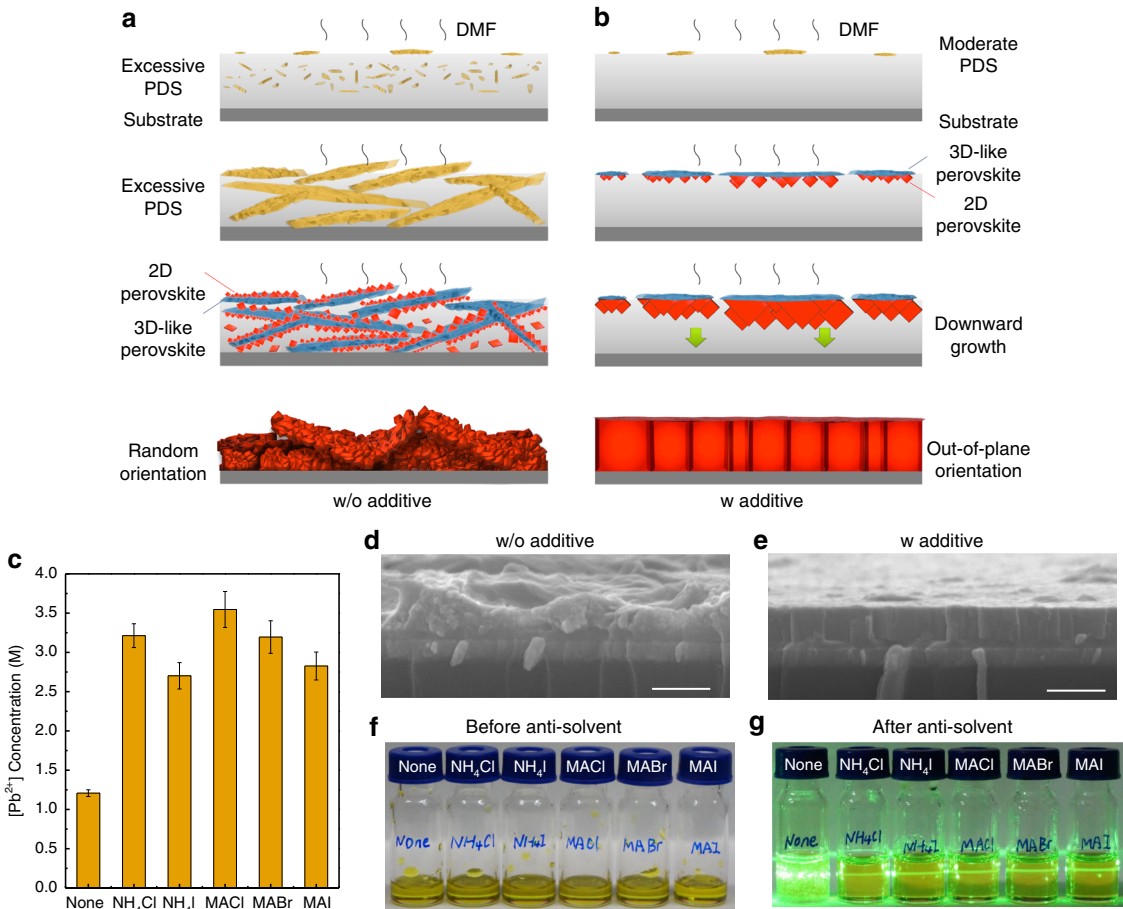

**Fig. 3 Competition of crystal growth models and suppressed nucleation by AX salts. a**, **b** Illustration of the 3D-like perovskites defined nucleation and growth of RP crystals from precursor solution without (**a**) or with (**b**) excessive AX additives; **c** influence of the AX additives (AX/PbI$_2$ = 0.5) on the critical concentration of precursor solution for the nucleation of PbI$_2$-DMF solvated phase in DMF, the error bars represent the s.d. of three measurements; **d**, **e** cross-sectional SEM images of BA-based RP perovskite films spun from precursor solution (<$n$> = 4) without (**d**) or with (**e**) NH$_4$Cl additives. The scale bar is 500 nm; **f** photos of BA-based RP perovskite precursor solutions (<$n$> = 4, 210 μL) without additive or with AX additives (AX/PbI$_2$ = 0.5) before the injection of chlorobenzene; **g** precursor solutions after injecting 540 μL CB as antisolvent, in which the precipitation of PDS was suppressed in the solutions with AX additives, as indicated by Tyndall effect.

(Supplementary Fig. 11 and Supplementary Note 6) revealed that the OP orientation and high crystallinity of the RP perovskites can be equally achieved in a group of films with excessive AX salts (A = NH$_4^+$ or MA$^+$; X = Cl$^-$, Br$^-$ or I$^-$) as additives. A common feature for precursors with excessive AX salts we have proved here is the suppressed precipitation of PDS in solution as identified by Tyndall effect in Fig. 3f, g (Supplementary Note 5), which is attributed to the enhanced solubility of PbI$_6$ octahedral colloids by AX salts (Fig. 3c).

To understand why layered perovskites prefer to grow from 3D-like perovskites coated PDS with a preferred orientation, we look into the crystallographic structures by examining the lattice matching between them. It can be noticed that the lattice matching between the facets of 3D-like perovskites and layered perovskite may play the key role in defining the crystal orientation. For the preformed PDS microcrystals with one-dimensional structure, lying horizontally with length direction parallel to the liquid-air interface are energetically preferred (Fig. 4a). Meanwhile, the [100] direction of the double (or triple) chain of edge-sharing Pb-I based octahedra is actually along the length direction of the PDS crystal (Fig. 4b) with a lattice constant of ~4.53 Å[30]. After the intercalation of MAI, the edge-sharing Pb-I based octahedra chains rotate to form corner-sharing PbI$_6$ octahedra networks (i.e., 3D-like perovskites) with abundant

periodic I$^-$ ions separated by ~6.3 Å along its length direction (see more details in Fig. 4c, Supplementary Fig. 12, Supplementary Note 7 and Supplementary Movie 1). Similar octahedra network rotation process has been proved previously[33,34], where the layered trigonal PbI$_2$ with (0 0 1) plane parallel to substrate can be converted into a 3D tetragonal MAPbI$_3$ with (1 1 0) plane parallel to substrate, which is analogous to the orientation evolution of octahedra shown in Fig. 4c. The lattice constant of I$^-$ ions along the length direction of 3D-like perovskites matches well with those periodic I$^-$ ions along [10$\bar{1}$] direction of RP perovskites with a spacing of 6.32 Å (Fig. 4d). Among these I$^-$ ions, the chains of I$^-$ ions located at those exposed corners of 3D-like perovskites and RP crystal sheet are reactive low-coordinated ions because those I$^-$ ions only form one Pb-I bond with adjacent Pb$^{2+}$ ion. The sharing of these low-coordinated I$^-$ ions chains between 3D-like perovskites and RP perovskites can significantly lower the energy barrier required for the nucleation, and template the alignment of RP perovskite crystal sheets, which is hence termed templated growth here. Another fact is that those low-coordinated I$^-$ ion chains on RP perovskite sheets along [1 0 0] or [0 0 1] directions (i.e., forming an angle of 45° with [10$\bar{1}$] and [1 0 1] direction) have different spacing (~4.47 Å, Fig. 4d). As a result, the (1 n 1) planes of RP perovskites are the most geometrically favorable candidate planes to attach with corner-

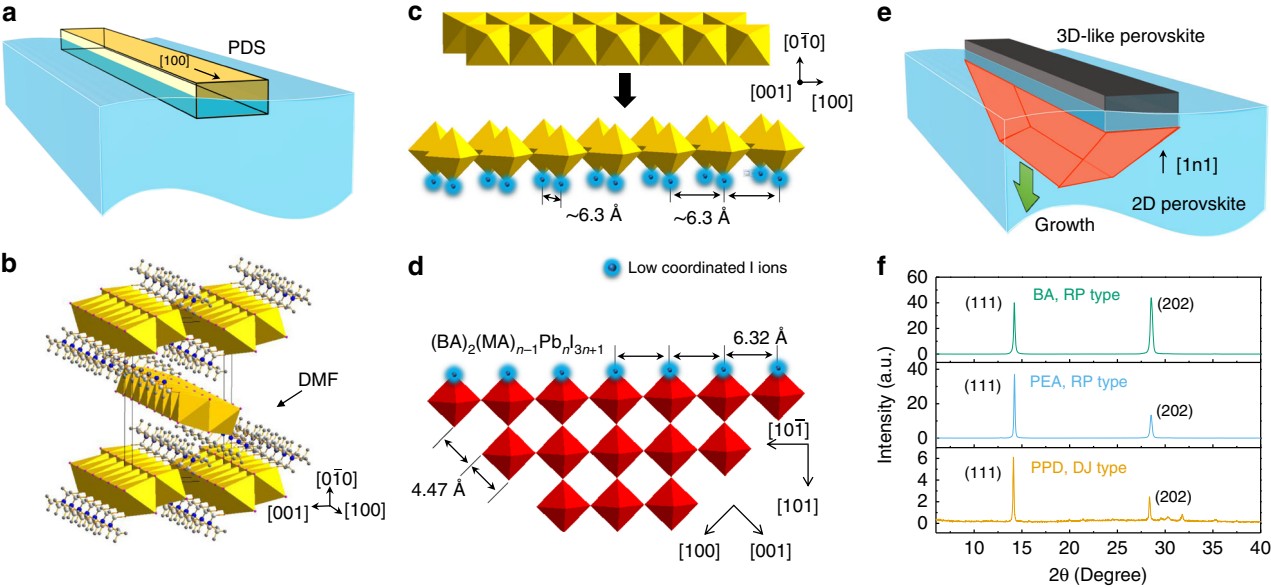

**Fig. 4 Lattice-matching defined crystal orientation. a** Illustration of the one-dimensional PDS crystals lying on solution surface; **b** the edge-sharing Pb-I octahedra chains in one-dimensional PDS crystal. **c** Illustration of the conversion of edge-sharing Pb-I octahedra chains into corner-sharing octahedral chains (see detailed process in supplementary movie 1), in which the I⁻ ions at the exposed corners are low-coordinated I⁻ ions (blue colored). **d** Illustration of the corner-sharing $PbI_6$ octahedra in the (0 1 0) plane of RP perovskites and the low-coordinated I⁻ ions (blue colored) at the edge of crystal sheet along the $[10\bar{1}]$ direction. **e** Illustration of the templated downward growing layered perovskite crystals on 3D-like perovskites, in which the dominating orientation is defined by the lattice matching shown in (**c**) and (**d**). **f** XRD spectra of BA-based RP perovskite film ($<n> = 4$), PEA-based RP perovskite film ($<n> = 4$), and PPD-based DJ perovskite film ($<n> = 4$) with OP orientation, in which the dominating diffraction peaks are all (1 1 1) and (2 0 2) peaks.

sharing $PbI_6$ octahedra chains in 3D-like perovskites for nucleation. As a consequence, when 3D-like perovskites triggers a downward growth of RP perovskites during solution thinning, the (1 0 1) or/and (1 1 1) planes of the RP crystal should be parallel to the substrate (Fig. 4e), which well explains the widely observed (2 0 2) and (1 1 1) XRD diffraction peaks and the absence of (n 0 0) and (0 0 n) peaks in RP perovskite film samples with OP orientation (Fig. 4f)[1,3,23,31,35]. The orientation of RP crystal in Fig. 2h also agree with the proposed templated growth behavior, in which the planes of RP perovskite forming an angle of 45° with PDS surface is assigned to the (n 0 0) and (0 0 n) planes; and meanwhile the (1 0 1) plane in RP perovskite, forming angles of 45° with (n 0 0) and (0 0 n) planes, is the plane that connect with 3D-like perovskites.

Moreover, based on a similar principle, dominated OP orientation in phenylethylammonium (PEA) based RP type and p-phenylenediamine (PPD) based DJ type layered perovskites have been achieved in our study by using AX salts as additives (Supplementary Fig. 13 and Supplementary Note 8). The XRD spectra of these OP orientated layered perovskite are also dominated by (1 1 1) and (2 0 2) diffraction peaks (Fig. 4f), which can be explained by a similarly templated growth behavior because the lattice constant along $[10\bar{1}]$ direction of PEA-based RP type and PPD-based DJ type layered perovskites are also ~6.3 Å. This agreement in crystal orientation further suggests the universality of this templated growth mechanism.

For clarification, we emphasize that those I⁻ ions on the edge-sharing $PbI_6$ are less active in the templated growth of RP perovskites. For example, soaking pure $PbI_2$ single crystal plate, with most of its I⁻ ions bonded with three adjacent $Pb^{2+}$ ions, into oversaturated RP precursor solution do not trigger templated growth of RP perovskites at the same condition (Supplementary Fig. 14), demonstrating the key role of the low-coordinated I⁻ ions on corner-sharing $PbI_6$ octahedra chains. On the other hand, $MA^+$ ions have been confirmed to be important in facilitating the templated growth. As mentioned above, it is difficult to achieve

OP orientation in RP perovskites with $n = 1$[3,14]. The losing of OP orientation in $BA_2PbI_4$ ($n = 1$) perovskites is also observed in our study with $NH_4Cl$ as additive (Supplementary Fig. 15). Coincidently, we found the 2D $BA_2PbI_4$ crystal sheets ($n = 1$) are not stable on the $PbI_2$-DMF surface. To further demonstrate, dipping $PbI_2$-DMF fibers into oversaturated RP precursor solution with $n = 1$ (i.e., no $MA^+$ ions) only lead to all the formed $BA_2PbI_4$ crystal fragments peeling off from PDS, finalized with disordered orientation (Supplementary Fig. 16). Due to the lack of $MA^+$ ions intercalated 3D-like perovskites phase, directional growth of $BA_2PbI_4$ crystal from $PbI_2$-DMF is unfavorable, which explains the loss of OP orientation in $BA_2PbI_4$ ($n = 1$) perovskites.

**OP orientation engineering for high performance devices.** Understanding the nucleation and directional growth of RP perovskites highlights the importance of engineering the formation process of solvate phase in the early stage of the solution thinning, which opens ways to manipulate crystal orientation and film morphologies in different fabrication techniques. For spin-coated RP perovskites solar cells (RPSCs), p-i-n structure was focused on (Fig. 5a). We select $NH_4Cl$ as the primary additives, because it yields the highest PCE. The impact of $NH_4Cl$ additive on the crystallinity of RP films has been measured with X-ray diffraction (XRD, Supplementary Fig. 17). The much narrowed full width at half maximum (FWHM) of (1 1 1) peaks and the absence of diffraction peaks below 10° further confirmed the high degree of crystallinity and dominating OP orientation induced by $NH_4Cl$ additive[1]. Accordingly, the PCE of $<n> = 4$ RP perovskite solar cells was dramatically improved from less than 1 to 13.2% (Fig. 5b) after $NH_4Cl$ addition with room temperature (RT) precursor solution and substrate. For RPSCs with $<n> = 5$, a high PCE of 14.4% was achieved, which is the highest reported efficiencies for BA-based RP perovskite solar cells fabricated with RT method (Fig. 5b, Supplementary Fig. 18, Supplementary Tables 1, 2 and Supplementary Note 9)[1,2,23]. In our study, the RPSCs

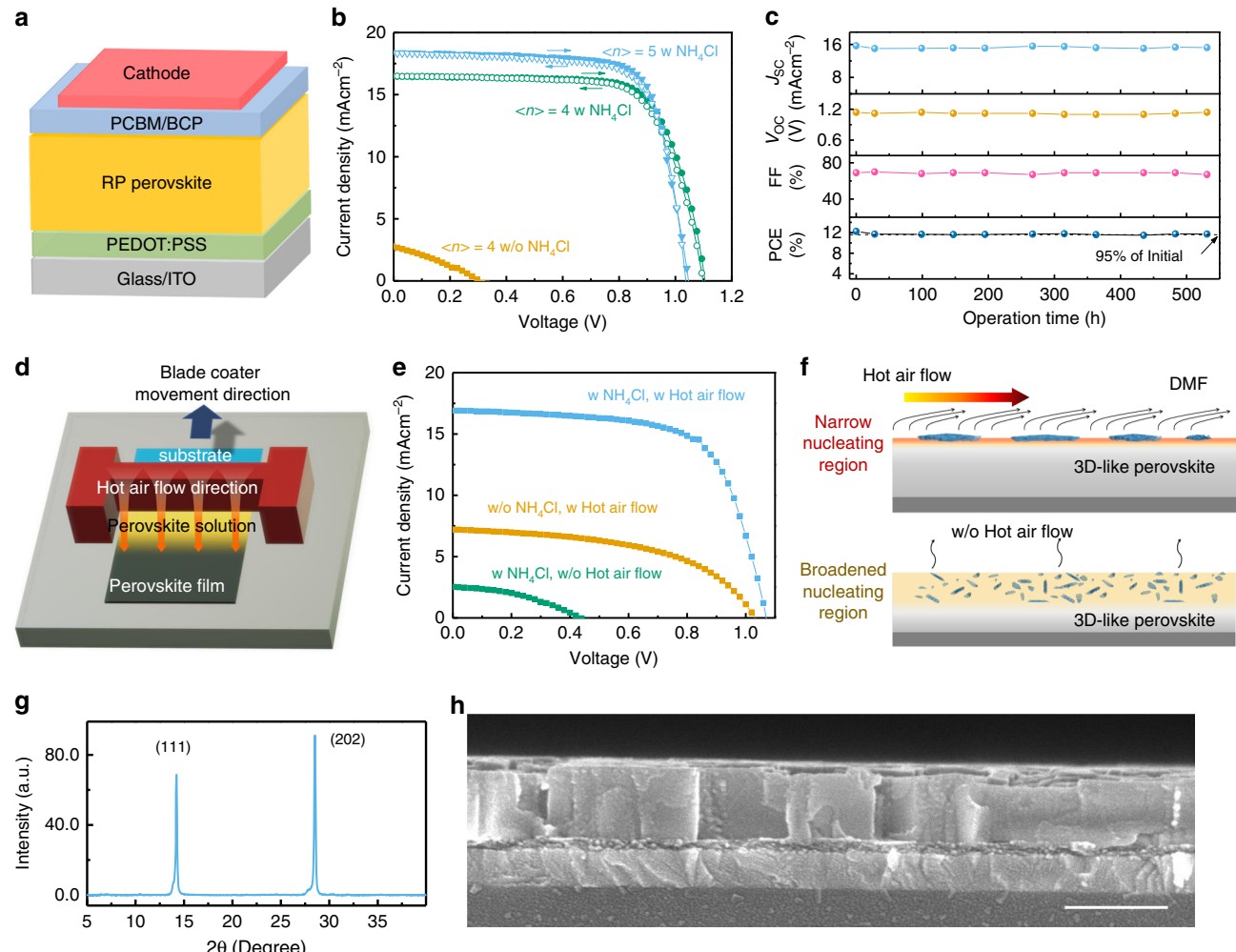

**Fig. 5 Achieving OP orientation in doctor bladed RP perovskite films. a** Scheme of the structure of RPSCs. **b** Comparison of current density ($J$)-voltage ($V$) curves of RPSCs (under 100 mW cm$^{-2}$ AM1.5 G illumination) without and with NH$_4$Cl additive, in which RPSCs with $<n>$ = 4 or 5 (with NH$_4$Cl) are shown. **c** Output parameters of BA-based RPSCs as a function of continuous operation time (aged under one-sun, 100 mW cm$^{-2}$, with a loading resistance of 1000 ohm); **d** illustration of doctor blading setup with RT substrate, precursor solution and available hot air flow. **e** Comparison of $J$-$V$ curves of RPSCs fabricated from doctor blading with and without NH$_4$Cl additive and hot air flow. **f** Illustration of the hot air flow induced narrow 3D-like perovskites formation region on the liquid surface and the slow evaporation (w/o hot air flow) induced 3D-like perovskites dispersion in a broadened region inside the precursor solution. **g**, **h** XRD spectrum (**g**) and cross-sectional SEM image (**h**) of doctor bladed RP films ($<n>$ = 4) with NH$_4$Cl additive and hot air flow co-treatment on ITO/PEDOT:PSS surface. The scale bar is 300 nm.

fabricated with NH$_4$Cl additives from spin-coating method are stable during operation at maximum power output point. Figure 5c shows our encapsulated RP perovskite solar cells ($<n>$ = 4) working maximum power point at can maintain 95% of its initial PCE value (12.3%) after 500 h of continuous operation (in air, one-sun, 100 mW cm$^{-2}$).

As another demonstration, solvate phase engineering has been applied to doctor bladed RP perovskite solar cells ($<n>$ = 4) to trigger OP orientation (Fig. 5d, e). During the doctor blading process, NH$_4$Cl was used to suppress the nucleation in the bulk of the precursor solution. However, solely using NH$_4$Cl additive do not guarantee OP orientation in RP perovskite film. Since the volatilization of DMF at RT in the doctor bladed process is much slower than that in the spin-coating process, the precipitation of PDS at the liquid surface is not predominant since the diffusion of solution and PDS is taking place (Fig. 5f). According to the understanding provided by this study, methods that facilitate 3D-like perovskites formed on top of the precursor solution is important for OP orientation. Based on this designing idea, hot air flow was employed to accelerate DMF volatilization on the

liquid surface with the solution and substrate unheated, which quickly form narrow oversaturated region located at the liquid surface (Fig. 5f). As a consequence, a dominating OP orientation was achieved in doctor bladed RP layered perovskites as verified by XRD and cross-sectional SEM studies (Fig. 5g, h). This NH$_4$Cl additive and hot air flow co-treatment resulted in vertical carrier mobility of ~0.24 cm$^2$ V$^{-1}$ s$^{-1}$, PL lifetime of ~48 ns and hence high PCE of 12.2% (Supplementary Fig. 19, Supplementary Note 10 and Supplementary Table 3). Due to the solubility difference shown in Fig. 2b, there should be a preferably precipitation of PbI$_2$-DMF and MAI-PbI$_2$-DMF solvated phase on top and BA-rich phase on bottom during spin coating or doctor blading, which leads to relative more larger-n RP perovskites formed on top of the resulted film, contributing to the frequently observed vertical phase separation (e.g., see the PL study of our sample in Supplementary Fig. 20)[36,37].

In conclusion, the bonding effect of low-coordinated I$^-$ ions on corner-sharing PbI$_6$ octahedra chains of 3D-like perovskites is figured out to be a strong driving force to trigger the nucleation of layered perovskites. The lattice matching between layered

perovskite and corner-sharing $PbI_6$ octahedra chains makes the orientation of layered perovskite substantially defined by the preformed 3D-like perovskites in solution. These insights offer general guidance to manipulate the crystal nucleation and film morphology in different solution fabrication processes (e.g. doctor blading) by means of solubility engineering and solution-drying engineering. Moreover, the thermodynamically available templated growth of layered perovskites can be used to construct heterojunction structure based on low dimensional perovskite crystals with different orientation, which would open up avenues to achieve perovskite optoelectronic devices with functional nanostructures.

## Methods

**Materials**. N, N-dimethylformamide (DMF, 99.8%), chlorobenzene (CB, 99.8%), $PbI_2$ (99.999%), $NH_4Cl$ (99.5%), $NH_4I$ (99.999%), n-butylamine (BA) phenylethylamine (PEA), p-phenylenediamine (PPD) and bathocuproine (BCP, 99.99%) were purchased from Sigma-Aldrich. Methylammonium iodide (MAI), methylammonium bromide (MABr) and methylammonium chloride (MACl) were purchased from Greatcell Solar Ltd. Poly(3,4-ethylenedioxythiophene): poly(styrenesulfonate) (PEDOT: PSS) AL4083 was purchased from Heraeus Ltd. [6,6]-phenyl-C61-butyric acid methyl ester (PCBM) was purchased from Solenne BV. Hydroiodic acid (HI, 55.0–58.0%) was purchased from Aladdin. All reagents and solvents were used directly if not specified.

**Solution preparation and device fabrication**. PEDOT: PSS was spin-coated on pre-cleaned indium tin oxide (ITO) substrate at 3000 rpm for 40 s and annealed at 125 °C for 20 min in air. RP perovskite precursor solution was spin-coated in glove box with $N_2$ or blade-coated in air. For RP perovskite precursor solution, MAI and $PbI_2$ were separately dissolved in DMF with the concentration of 500 mg mL$^{-1}$. Then, $BA_2MA_{n-1}Pb_nI_{3n+1}$ RP perovskite solution was prepared by mixing BA: MAI:$PbI_2$ with a molar ratio of $2:n + 1:n$. Additives AX (A = $NH_4^+$ or $MA^+$; X = $Cl^-$, $Br^-$ or $I^-$) was first dissolved in $PbI_2$/DMF solution at 65 °C, then MAI and BA were mixed before adding into $PbI_2$ + AX solution. Both precursor solution and substrates were kept at room temperature during deposition. The obtained RP perovskite films were annealed at 65 °C for 5 min and 100 °C for 30 min for better crystallinity and to remove the $NH_4Cl$ additives. PCBM with a concentration of 15 mg mL$^{-1}$ in CB was spin coated on perovskite at 3000 rpm for 30 s and annealed at 80 °C for 10 min. At last, BCP (7 nm thick) and Cu (80 nm thick) were evaporated sequentially on the films in vacuum at a rate of 0.2 Å s$^{-1}$ and 2 Å s$^{-1}$, respectively. The device area is defined to be 0.10 cm$^2$ by metal masks.

**Templated growth of RP perovskites**. Oversaturated BA-based RP perovskite ($<n> = 2$) aqueous solution was prepared by dissolving 76.84 mg MAI, 53 mg $PbI_2$ and 32.9 μL BA into 800 μL HI solution at 80 °C. The oversaturated solution was obtained after cooling the solution to 55 °C. Then, the PDS phase (both $PbI_2$-DMF and MAI-$PbI_2$-DMF solvated phases) was introduced into the oversaturated solution for the growth of RP perovskites. For intentionally accelerating the solidifying process of RP perovskite in CB, oversaturated BA-based RP perovskite ($<n> = 4$) DMF solution was prepared by dissolving 461 mg $PbI_2$, 198.8 mg MAI, 26.8 mg $NH_4Cl$ and 49.4 μL BA in 230 μL DMF.

**Characterization**. The current density ($J$)-voltage ($V$) curves of RPSCs were measured in nitrogen glove box by Keithley 2400 with a voltage scan rate of 0.02 V s$^{-1}$, delay time of 50 ms and sweep region from −0.2 to 1.2 V under 100 mW cm$^{-2}$ AM 1.5 G illumination provided by an AAA class solar simulator (Enli Technology Co., Ltd.). A NREL certified Si reference cell (SRC-2020, Enli Technology Co., Ltd) was used for calibration. The external quantum efficiency (EQE) was characterized by the QE-R solar cell quantum efficiency measurement system (Enli Technology Co., Ltd., China), and the light source is a 75 W xenon lamp. The monochromatic light intensity for EQE was calibrated with a NIST-certified Si photodiode from 300 to 1100 nm. The EQE spectrum was integrated over AM 1.5 G photon flux to attain photocurrent density. XRD measurements were carried out in air using a Siemens D500 Bruker X-ray diffractometer (Cu Kα radiation, $\lambda$ = 1.5406 Å). SEM images of the perovskite crystals are obtained by using a scanning electron microscope (TESCAN MIRA3 LMU) equipped with an electron beam accelerated at 10–20 kV. The EDS was measured by X-Max20 silicon drift detector (Oxford). The absorbance spectra were obtained by using a UV–Visible Spectrometer (Thermo Evolution 201) in the spectral range of 300–1100 nm. Steady-state PL was measured by i-HR320 spectrometer (HORIBA Scientific) with excitation by a UV laser (337 nm). The spatial resolved infrared spectra were measured by nanoscale IR spectroscopy (nanoIR2, Bruker) with a lateral spatial resolution of 100 nm. Fourier transform infrared (FTIR) was measured by Nicolet iS50 (Thermo). GI-XRD measurement was performed on a Xenocs Xeuss 2.0 system. The wavelength of the X-ray beam is 0.154 nm with a flux of approximately $4.6 \times 10^7$ photons s$^{-1}$ and an illumination area of $1.2 \times 1.2$ mm$^2$. The incident angle of

the X-ray beam was set as 0.5°. The 2D GI-XRD patterns were collected by a Pilatus 300K detector. The sample to detector distance was 170 mm, calibrated by the silver behenate standard sample. The GI-XRD patterns were analyzed using the software package FIT2D.

## Data availability

The data that support the findings of this study are available from the corresponding author upon reasonable request.

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

## Acknowledgements

We thank the financial support from National Natural Science Foundation of China (51673218, U1632265, 51802194, 61774170, and 61874141). J. Huang thanks financial support from UNC Research Opportunities Initiative (ROI) through the Center of Hybrid Materials Enabled Electronic Technology. Yuan thanks the Innovation-Driven Project of Central South University, the open Fund of the State Key Laboratory of Integrated Optoelectronics (IOSKL2016KF05), the financial support from the State Key Laboratory of Powder Metallurgy at Central South University. Luo thanks the Central South University postdoctoral international exchange introduction program, China Postdoctoral Science Foundation.

## Author contributions

Y.Y. conducted the project. Y.Y. and J. Huang conceived the idea, designed the experiments, analysis the data and wrote the paper. J.W. fabricated all spin-coated RP films and solar cells, and carried out the J-V curves and EQEs measurement. J.W. and S.L. carried out the XRD, SEM, Abs, PL characterizations. S.L. carried out the nucleation concentration measurement, antisolvent experiment and crystallography analysis. Z.L., K.M. and G.C. carried out GIWAXS measurement. C.Z. and L.D. took part in analyzing the XRD and GIWAXS results. Y.C. carried out templated grain growth of RP perovskite on PDS. Y.L. fabricated RPSCs by doctor-blading and the corresponding characterization. T.H. and S.L. carried out the IR-AFM measurement. Y.L., J. He and X.S. contributed to TRPL and H.H. contributed to AFM studies. Y.D. carried out optical study on bladed film. All the authors revised the paper.

## Competing interests

The authors declare no competing interests.
