## [Peer Review File · Nature Communications]

Reviewers' comments:

Reviewer #1 (Remarks to the Author):

The work titled “Templated Growth of Oriented Layered Hybrid Perovskites on Solvated Perovskites” aims to study the mechanism of 2D perovskites growth and to understand how the orientation of layered perovskites is controlled with different treatments.

The authors first study the effect of NH₄Cl on the orientation of 2D perovskite thin films. Then the authors investigate the growth of single crystals and the effect of a PbI₂-DMF solvate phase on the growth and orientation of the crystals, and strive to connect the observations to the mechanisms that control 2D perovskite orientation in perovskite films.

Finally, the authors fabricate solar cells with the goal of corroborating their findings.

This reviewer found that there were many interesting observations in this work. However, there are also oversights, unwarranted assumptions, and a lack of flow throughout the manuscript. In its present state the manuscript cannot be accepted for publication in Nature Communications. With the appropriate major revisions, however, it might become suitable.

Here are major items requiring attention:

1. The authors give no motivation for studying NH₄Cl treatment. Has this been studied before? If so, why are you studying it again? The NH₄Cl work seems very disjointed relative to the rest of the manuscript and there is no convincing argument for the connection between this and the PbI₂-DMF solvate work. If the authors insist on including the NH₄Cl work it must be introduced, motivated, and connected with the rest of the work.
2. A lack of low-*q* XRD and GIWAXS peaks does not prove OP orientation. The authors are assuming that the both the control and NH₄Cl-treated films exhibit the same phase of 2D perovskite. This may not be true. The NH₄Cl could be producing 2D perovskites of higher *n*-values, or even bulk perovskite. The authors must perform measurements to verify that these films are made of the same *n*-valued 2D perovskite. Optical measurements offer the best characterization for this. The authors should at least perform absorption and photoluminescence spectroscopy. Transient absorption spectroscopy would also be quite useful.
3. The labelling of XRD peaks is not unequivocal, and at times is fairly unconventional. Bulk perovskite 110 and 002 peaks overlap near the $q=10 \text{ nm}^{-1}$ part of the spectrum. The labels used could be one possible labelling option, but can also be explained by a perpendicular basis. With this point and the previous in mind, the GIWAXS in Figure 1 does not prove to this reviewer that NH₄Cl

leads to OP 2D phases. NH_4Cl may just hinder the growth of 2D phases, and lead to very strongly oriented bulk perovskite. On a related note, some of the labels in the GIWAXS figures have 4 numbers (2102) and (0102). 4 numbers are sometimes used in hexagonal bases, what does this mean in a cubic basis? The authors should stick to the more conventional 3-number basis set.

The language throughout the manuscript is more definitive than is justified by the evidence/arguments. Some of the observations made may support the hypothesis of the authors, but it is not absolute evidence for it. The authors should point to any evidence that could support their claims instead of making unsubstantiated claims. I list some examples:

The authors claim that the 750 nm absorption is evidence for a solvated perovskite? What is the evidence for this? This could also be bulk perovskite. DMF is known to evaporate fairly easily at room temperature.

What is the evidence that the phase observed in SEM is $\text{PbI}_2\text{-DMF}$? It could also be $\text{MAI-PbI}_2\text{-DMF}$. And that goes for the rest of the manuscript, could $\text{MAI-PbI}_2\text{-DMF}$ be participating instead of $\text{PbI}_2\text{-DMF}$?

The authors state that NH_4Cl suppresses the $\text{PbI}_2\text{-DMF}$ phase. Is there any evidence for this?

The authors state that lattice-matching defines orientation. The authors do not have the evidence to make such strong claims.

The authors state that the lowest energy configuration between the PbI_2 -phase and liquid is along the long side. Again, what evidence is there for this?

Overall, the writing can be substantially improved. There are a number of instances of clerical errors, poor word choice, and informal language (e.g., “a bunch of”, “it turns out”, “warranties”).

Reviewer #2 (Remarks to the Author):

In this manuscript, Wang et al. report the templated growth of layered perovskites with out-of-plane orientation on solvated perovskites. The growth mechanism demonstrated in this work is reasonably proved by experimental data, which provides insightful understanding on the growth of vertically oriented layered perovskite films using perovskite precursor solutions with additional ammonium halide salts. In addition, the presented RP solar cells show device performance comparable to the best of the reported layered solar cells. The manuscript is well written and would be of great interest to the perovskite community. Therefore, I strongly recommend the publication of this work in Nature Communications. The following comments are provided to the authors to further strengthen the manuscript.

1. When soaked in oversaturated RP precursor solution, the surface of PDS powders turn black in a few seconds (Supplementary Figure 5). Then the surface of the PDS induces the growth of oriented RP perovskites. However, it is unclear when the PDS totally turns into perovskite phase. Does it

occur during solution thinning, after spin-coating or during thermal annealing?

2. The PDS (PbI₂-DMF solvated phase) plays a critical role in the growth of oriented layered perovskites. Could the DMF in PDS be other solvent molecules, e.g. DMSO?

3. Ammonium halide salts such as NH₄Cl improve the solubility of PDS, which has been proved by the authors. Could the authors further explain why these salts can improve the solubility of PDS?

4. As reported in literature, e.g. *Adv. Energy Mater.* 2018, 8 (21), 1800185 and *J. Am. Chem. Soc.* 2017, 139, 1432-1435, RP perovskite films prepared from nominal n values are actually mixtures with various RP phases. These RP perovskite films show graded phase distribution even with out-of-plane orientation. Supplementary Figure 17 in this work also shows a similar observation. Could the authors comment on the growth mechanism of graded RP phases with out-of-plane orientation?

Reviewer #3 (Remarks to the Author):

The manuscript by Wang et al. presents a thorough investigation of the growth mechanism for oriented layered perovskites. The main finding is that the firstly formed solvated perovskites in solution can template the sequential growth of layered perovskites, the orientation of layered perovskites is determined by the lattice matching between solvated perovskite and layered perovskites. In my opinion, since H. Tsai et al. reported hot-cast RP-type layered perovskites with out-of-plane orientation (*Nature* 536, 312, (2016)), the detailed nucleation process of layered perovskites is still unclear. The finding in this manuscript is very exciting to the 2D perovskite field, because the authors clarified several unsolved questions with this templated growth mechanism, especially how the out-of-plane orientation was formed. Moreover, it was well demonstrated that the formation of solvated perovskites can be engineered, which could be a useful method to achieve layered perovskite films with out-of-plane orientation and hence improve device performance. The manuscript is well written and organized with ample relevant references. I consider this work to be suitable for the broad readership in *Nature Communications* and I would recommend publication of this manuscript after revision.

The following are some specific concerns:

1) According to some previous results (e.g. K. Yan et al. *J. Am. Chem. Soc.* 137, 4460 (2015)), the perovskite precursor solutions are generally colloidal dispersions, so I am wondering whether the excessive NH₄Cl also impacted on the colloidal size and hence tuned the grain size and final morphology of the layered perovskite films?

2) It is impressive that the out-of-plane orientation and high crystallinity of the RP perovskites can be equally achieved in a group of films with excessive AX salts (A = NH₄⁺ or MA⁺; X = Cl⁻, Br⁻ or I⁻) as additives. So how about the device performance for solar cells resulted from precursor solutions with other AX salts than NH₄Cl?

3) In Fig. S14a, the XRD pattern of RP films with different amounts of NH₄Cl addition was shown, the absence of diffraction peaks below 10° when the molar ratio of NH₄Cl: PbI₂>0.3 was believed to be due to the dominating OP orientation induced by NH₄Cl additive. However, there is an exception for 1.0 NH₄Cl, in which some extra diffraction peaks show up. Could the authors provide some discussion on this?

Response to Referees Letter

Referee #1:

Comment 1: *The work titled “Templated Growth of Oriented Layered Hybrid Perovskites on Solvated Perovskites” aims to study the mechanism of 2D perovskites growth and to understand how the orientation of layered perovskites is controlled with different treatments. The authors first study the effect of NH_4Cl on the orientation of 2D perovskite thin films. Then the authors investigate the growth of single crystals and the effect of a PbI_2 -DMF solvate phase on the growth and orientation of the crystals, and strive to connect the observations to the mechanisms that control 2D perovskite orientation in perovskite films. Finally, the authors fabricate solar cells with the goal of corroborating their findings.*

This reviewer found that there were many interesting observations in this work. However, there are also oversights, unwarranted assumptions, and a lack of flow throughout the manuscript. In its present state the manuscript cannot be accepted for publication in Nature Communications. With the appropriate major revisions, however, it might become suitable.

Here are major items requiring attention: The authors give no motivation for studying NH_4Cl treatment. Has this been studied before? If so, why are you studying it again? The NH_4Cl work seems very disjointed relative to the rest of the manuscript and there is no convincing argument for the connection between this and the PbI_2 -DMF solvate work. If the authors insist on including the NH_4Cl work it must be introduced, motivated, and connected with the rest of the work.

Response 1: We appreciate the referee for the insightful comments which helped us to improve the quality of this manuscript tremendously. We also thank the referee’s recognition of the novelty of this work, and think this manuscript can be accepted after revisions. We followed the referee’s suggestion to enhance our proving as will be shown in the following point-by-point responses.

The major contribution of this work is to understand the nucleation and alignment of layer perovskites. We revealed that, in order to make layered perovskite grow directionally, a surface initialized crystallization is needed, which only become dominating when the nucleation inside the solution is suppressed and so that nucleation starts at liquid surface. Due to this reason, NH_4Cl additive is necessary in our work. As we mentioned in the manuscript and will discuss below in this response, NH_4Cl additive played an important role in the suppression of the precipitation of PbI_2 -DMF and MAI- PbI_2 -DMF solvated phase phases, which guarantee the downward growth of RP perovskite from liquid surface as dominating mechanism. Without employing NH_4Cl as additive, the templated growth of RP perovskite from quasi-3D perovskite can still happen, but sole templated growth mechanism does not result in RP perovskite films with high crystallinity and well aligned crystal orientations at RT. In another word, we agree that employing NH_4Cl additives has been reported previously mainly in MAPbI_3 -based 3D perovskites (e.g. Ding, L. and et al., *Nanoscale* 6, 9935-9938 (2014); Chen, T. and et al., *J. Mater. Chem. A* 3, 18514-18520 (2015); Liang, Z. and et al.,

Chem. Mater. 27, 1448-1451 (2015)). However, to the best of our knowledge, the influence of the NH_4Cl additives on the nucleation of PbI_2 -based solvated phase in solution has never been noticed and discussed before. We firstly discovered the role of NH_4Cl additive in this study. Since the nucleation engineering is crucial for the control of the final morphology of RP perovskites, we further broadened the idea that using additives like NH_4Cl , MgCl_2 , MAI , MABr or NH_4I can be general ways to engineer the nucleation of solvated phase and hence the RP perovskites.

Figure 5c | Output parameters of BA-based RPSCs as a function of continuous operation time (aged under one-sun, $100 \text{ mW}/\text{cm}^2$, with a loading resistance of 1000 ohm);

Supplementary Figure 18a | Distribution of reported PCEs of BA-based RPPCs ($\langle n \rangle = 3-5$) fabricated by room-temperature methods or hot-casting method, respectively.

It's worthy to mention that our study also demonstrated the good working stability (Fig. 5c) of the

RP perovskite solar cells fabricated with NH_4Cl additive method, which has never been discussed before, to the best of our knowledge. On the other hand, using NH_4Cl as additives is a very promising method for fabricating solar cells with comparable high PCE (see Supplementary Figure S18 of the revised SI) with that of hot casting method (Tsai, H. et al. Nature 536, 312-316 (2016); Zhang, X. et al. Energ. Environ. Sci. 10, 2095-2102 (2017)), which makes our conclusion representative.

So logically, introducing NH_4Cl treatment is necessary, which is closely connected with other analysis in our manuscript. We prefer to remain it in our manuscript. As the referee pointed out, we missed the motivation of using NH_4Cl as additive in our previous manuscript. We accepted this suggestion and made some revises as follows:

In page 4, last paragraph, we add a sentence: “As inspired by the success of NH_4Cl additive in promoting the crystallinity of 3D perovskites, recently NH_4Cl additive was also employed in RP perovskite for grains with OP orientation”.

In page 5, last paragraph, we add Ref.23 and Ref. 24 after the sentence of “The changing of dominating crystal orientation from IP to OP by NH_4Cl additive offers an excellent platform for in-depth investigation of the nucleation and growth of RP perovskites”

Comment 2: *A lack of low- q XRD and GIWAXS peaks does not prove OP orientation. The authors are assuming that the both the control and NH_4Cl -treated films exhibit the same phase of 2D perovskite. This may not be true. The NH_4Cl could be producing 2D perovskites of higher n -values, or even bulk perovskite. The authors must perform measurements to verify that these films are made of the same n -valued 2D perovskite. Optical measurements offer the best characterization for this. The authors should at least perform absorption and photoluminescence spectroscopy. Transient absorption spectroscopy would also be quite useful.*

Response 2: We thank referee for the comments and suggestions.

By following the referee’s suggestion, we compared the absorption spectra of BA-based layered perovskite films fabricated from precursor solution with and without NH_4Cl , as it is shown in Supplementary Figure 1 of revised SI file. The absorption spectra of sample with NH_4Cl as additives show exciton absorption peaks located at 605, 640 and 668 nm (Supplementary Figure 1) which can be assigned to layered perovskite with $n = 3, 4$ and 5 , respectively. The absorption peaks agree well with other papers (Stoumpos, C. C. et al. Chem. Mater. 28, 2852-2867 (2016); Quintero-Bermudez, R. et al. Nat. Mater., 17, 900-907 (2018)). We can see sample with NH_4Cl additive show relatively higher content of layered perovskites phase with $n=3-5$. On the contrast, sample without NH_4Cl additive show higher content of layered perovskites phase with high n value (with absorption around 700~770 nm) and low n value phase (e.g. $n=2$ at 567 nm), but less content in $n=3-5$. We did not assuming that both the control and NH_4Cl treated film has the same phase of 2D perovskite in our manuscript. The result is that using NH_4Cl additive can suppress the nonuniform distribution of

layer number (n), other than increase the average layer number (because there were no NH_4^+ and Cl^- ions remained after annealing as we will prove it below).

Accordingly, in our revised Supplementary Information, we changed the Supplementary Figure 1 to the current version:

Supplementary Figure 1 | Absorption spectra of Ruddlesden-Popper (RP) perovskite films based on butylamine (BA) cation ($\langle n \rangle = 4$) fabricated from precursor solution with or without 0.5 molar ratio of NH_4Cl additives. Though the precursor solution was composed as $\text{BA}_2\text{MA}_3\text{Pb}_4\text{I}_{13}$ ($n=4$), the result RP perovskite film contained RP perovskites with different layer number n . Thus, we define average layer number $\langle n \rangle$ by the component in precursor solution.

For a better demonstration of that the photocurrent in layered perovskite (w NH_4Cl additive) is mainly contributed by low- n value (e.g. $n < 5$) phase (other than high- n value phase), we further compared the EQE spectra of perovskite solar cells based on layered perovskite ($\text{BA} \langle n \rangle = 4$) and 3D perovskite with the same solar cells structure (i.e. ITO/PEDOT:PSS/perovskite/PCBM/C60/BAPC/Cu) in Figure R1. The EQE drop dramatically to be less than 40% in the region of wavelength > 700 nm, which is significantly lower than that of MAPbI_3 based 3D perovskites. The lower EQE around 700-800 nm consistent with the EQE value ($< 40\%$) reported in Tsai et al.'s paper with $\langle n \rangle = 4$ (Tsai, H. et al. Nature 536, 312 (2016)) due to the lack of 3D perovskite phase. Meanwhile the EQE value is over 60% in the region of 340 nm \sim 640 nm, overlapping with that of MAPbI_3 based 3D perovskite due to the efficient light absorption of layered perovskite in this wavelength region. In the region of 780 nm \sim 800 nm, where only the MAPbI_3 perovskite phase contributed to the EQE signal, the EQE value of layered perovskite solar cells is only $\sim 1/8$ of that in pure MAPbI_3 based solar cells, indicating the very limited contribution of 3D perovskite phase in light harvesting and photocurrent in the RP layered perovskite solar cells. The EQE spectra further supports that the dominating

contributions in photocurrent are layered perovskites with low n value. We believe that the absorption and EQE spectra had provide enough evidence for the existence of low- n layered perovskite as dominating component, since the NH_4Cl additive does not remain in final layered perovskite films.

Figure R1 | EQE spectra of $\text{BA } \langle n \rangle = 4$ RP perovskite solar cells and MAPbI_3 perovskite solar cells with the same device structure of RP perovskite solar cells based on $\text{BA } \langle n \rangle = 4$ with NH_4Cl .

Supplementary Figure 20 | Normalized PL spectra of $\text{BA } \langle n \rangle = 4$ RP perovskite films with NH_4Cl additive under 337 nm laser illumination from perovskite side and glass side.

As for the PL measurement, somehow we don't think it can tell the amount of layered perovskite phase due to the presence of charge transfer or/and energy transfer process (Liu, J. et al. J. Am. Chem. Soc. 139, 1432-1435 (2017)). We have the photoluminescence spectra of $\text{BA } \langle n \rangle = 4$ RP perovskite films shown in Supplementary Figure 20. The existence of 2D perovskite can be

identified when light was incident from glass side. The PL is dominated by the peak at 750 nm, mainly contributed by materials phase with low band gap. So it's not straightforward for us to estimate the amount of layered perovskite with low- n value from photoluminescence spectra. We agree that transient absorption spectroscopy can provide more information in the phase distribution of 2D perovskite (Quintero-Bermudez, R. et al. *Nat. Mater.*, 17, 900-907 (2018)), but such facility is not accessible for us at this moment.

On the other hand, we want to mention that NH_4Cl do not increase the amount of 3D perovskite or high- n layered perovskite phases because it does not remained in the final perovskite film. From the tolerance factor point of view, the incorporation of NH_4^+ and Cl^- ions into the layered perovskites or 3D perovskite phase is thermal dynamically unstable due to their small ion radius (Colella, S. et al. *Chem. Mater.* 25, 4613-4618 (2013); Williams, S. T. et al., *ACS nano* 8, 10640-10654 (2014)). Moreover, in our studies, in order to further enhance the crystallinity of layered perovskites, we carried out thermal annealing under 100°C for 30 mins for all the samples. The NH_4^+ and Cl^- ions evaporate during the heating process, similar to the cases that already demonstrated in 3D perovskites (Si, H. et al., *Adv. Funct. Mater.* 27, 1701804 (2017); Xie, F. X. et al. *ACS Nano* 9, 639-646 (2015)). For further proof, we carried out XPS spectra analysis (see Figure R2 attached below), no Cl 2p core level peaks around binding energy of 200 eV can be found (e.g. Ng, T. et al. *J. Mater. Chem. A*, 3, 9081-9085 (2015)), excluding the presence of Cl element remained in the final perovskite films after annealing process.

Figure R2 | X-ray photoelectron spectroscopy (XPS) of BA based layered perovskite ($\langle n \rangle = 4$) fabricated from precursor solution with 0.5 molar ratio of NH_4Cl additive. The absence of Cl 2p core level peak around 200 eV suggests undetectable Cl^- ions remained in the perovskite film.

Comment 3: The labelling of XRD peaks is not unequivocal, and at times is fairly unconventional. Bulk perovskite 110 and 002 peaks overlap near the $q=10 \text{ nm}^{-1}$ part of the spectrum. The labels used could be one possible labelling option, but can also be explained by a perpendicular basis. With this point and the previous in mind, the GIWAXS in Figure 1 does not prove to this reviewer that NH_4Cl leads to OP 2D phases. NH_4Cl may just hinder the growth of 2D phases, and lead to very strongly oriented bulk perovskite. On a related note, some of the labels in the GIWAXS figures have 4 numbers (2102) and (0102). 4 numbers are sometimes used in hexagonal bases, what does this mean in a cubic basis? The authors should stick to the more conventional 3-number basis set.

Responses 3: Thanks for the comments and suggestions. As discussed in the response to comment 2, NH_4Cl additive does not increase the amount of bulk perovskite phase but reduce the amount of layered RP perovskite with large- n . We agree that the referee pointed out an important issue about how to prove the out-of-plane (OP) orientation of layered perovskites, which actually is a commonly faced problem. Currently we donot have better characterization method to directly observe the OP orientation of layered perovskites. As mentioned in response 2, we can prove the presence of significant amount of layered perovskite with $n=3-5$ by the absorption spectrum (Supplementary Figure 1) and EQE spectrum (Figure R2). Meanwhile, we did not find any evidence for the in-plane (IP) orientation of layered perovskite in those films. Many previous reports claimed OP orientation of RP perovskites by using GIWAXS pattern as their major evidence (Tsai, H. et al. Nature 536, 312-316 (2016); Chen, A. Z. et al. Nat. Commun. 9, 1336 (2018); Zhou, N. et al. J. Am. Chem. Soc. 140, 459 (2018); Lai, H. et al. J. Am. Chem. Soc. 140, 11639 (2018)); As compared in Figure R3, our GIWAXS patterns match well with those published works.

Figure R3 | Comparison of the GIWAXS patterns between figure extracted from: (a) Tsai's paper

(Tsai, H. et al. Nature 536, 312-316 (2016)), (b) Chen’s paper (Chen, A. Z. et al. Nat. Commun. 9, 1336 (2018)), (c) Zhou’s paper (Zhou, N. et al. J. Am. Chem. Soc. 140, 459 (2018)), (d) Lai’s paper (Lai, H. et al. J. Am. Chem. Soc. 140, 11639 (2018)), and (e) our manuscript. All the GIWAXS patterns show essentially the same diffraction peaks (e.g. the presence of (1 1 1), (2 0 2) peaks and the absence of diffraction peak along q_z axis in the range of $0\sim 10\text{ nm}^{-1}$), and the patterns in Figure R3a-d were used to prove the OP orientation of layered perovskites in the papers mentioned above, respectively.

On the other hand, we want to mention that, we did not find any evidence proved that adding NH_4Cl additive in MAPbI_3 based 3D perovskite can lead to “very strongly oriented bulk perovskite” as proposed by the referee. On the contrast, as reported by Chen et al. (Chen, Y. et al., Chem. Mater. 27, 1448 (2015)) and shown in Figure R4a, the obtained bulk perovskite film from precursor solution with NH_4Cl additive still has a broad distribution in crystal orientation of bulk perovskites, as indicated by the strong arcs in the GIWAXS pattern. In another case, as reported by Chen et al. (Chen, A. Z. et al. J. Mater. Chem. A 5, 7796-7800 (2017)), pristine 3D perovskite or 3D perovskite with NH_4Cl treatment do not form strongly orientated bulk perovskite grains. This can be seen in Figure R4b-d which is obtained from Figure S4 of Chen’s paper. So, despite of the similar position of the diffraction peaks of bulk perovskite along q_z direction with that of RP layered perovskites (i.e. around 10 nm^{-1}), the highly concentrated diffraction spots in our GIWAXS patterns are not likely caused by bulk perovskite phase because bulk perovskites with NH_4Cl as additive was reported to have more disordered crystal orientation.

Figure R4 | GIWAXS of MAPbI_3 bulk perovskite films. (a) 3D perovskite film with NH_4Cl as additive (Liang, Z. et al., Chem. Mater. 27, 1448-1451 (2015)) (b) pristine 3D perovskite after annealing; (c) 3D perovskite treated with NH_4Cl and before annealing; and (d) 3D perovskite treated with NH_4Cl and after annealing. (Chen, A. Z. et al. J. Mater. Chem. A, 5, 7796-7800 (2017))

In summarize, all the analysis mentioned above suggests that most of the layered perovskite adopted OP orientation, which is also consistent with the vertical carrier mobility of $\sim 0.24\text{ cm}^2/\text{Vs}$ and high FF of 73-75% (Table S2) obtained in our studies.

In page 5, the first paragraph in our revised manuscript, we modified the sentence “The absence of diffraction peaks along q_z axis in the range of $0\sim 10\text{ nm}^{-1}$ (Fig. 1b) confirms the

dominating OP orientations...” to be “The clear exciton absorption peaks for layered perovskite (Supplementary Figure 1a) together with the absence of diffraction peaks along q_z axis in the range of $0 \sim 10 \text{ nm}^{-1}$ (Fig. 1b) suggests a dominating OP orientations...”

As for the unequivocal and unconventional the XRD labelling, the 4 numbers labels (e.g. 2102 or 0102) were actually 3 number (i.e. (2 10 2) or (0 10 2)). In order to avoid misunderstanding, we added space in all the labels in our revised manuscript.

Comment 4: *The language throughout the manuscript is more definitive than is justified by the evidence/arguments. Some of the observations made may support the hypothesis of the authors, but it is not absolute evidence for it. The authors should point to any evidence that could support their claims instead of making unsubstantiated claims.*

I list some examples: The authors claim that the 750 nm absorption is evidence for a solvated perovskite? What is the evidence for this? This could also be bulk perovskite. DMF is known to evaporate fairly easily at room temperature.

Response 4: We agree with the referee that the black phase has a structure that is more close to bulk perovskite. In fact, we describe the black phase with a structure featured as corner-sharing PbI_6 octahedra networks (e.g. see Figure 4c), which is similar to 3D perovskites. We call it “solvated perovskite” since we concerned that DMF molecules can be dissolved in the corner-sharing PbI_6 octahedra networks (e.g. Xiao et al., Adv. Mater., 26, 6503–6509 (2014)). However, we found the term “solvated perovskite” might be confused with structures featured as edge-sharing PbI_6 octahedra networks, such as PbI_2 -DMF solvated phase (Wakamiya, A. et al. Chem. Lett. 43, 711-713 (2014)) or $(\text{MA})_2(\text{DMF})_2\text{Pb}_{n+2}$ phase (Petrov, A. A. et al. J. Phys. Chem. C 121, 20739-20743 (2017)).

In our revised manuscript, for a better accuracy, we changed the term of “solvated perovskite” to “3D-like perovskite” since the black color phase is similar to 3D perovskite. We use the term of “3D-like” because the corner-sharing PbI_6 octahedra networks (i.e. the black colored phase) is not complete as compared to ideal bulk perovskite due to the absorption of DMF molecules and less continuous in its octahedra network. Here we thank referee for the constructive comments.

Comment 5: *What is the evidence that the phase observed in SEM is PbI_2 -DMF? It could also be MAI- PbI_2 -DMF. And that goes for the rest of the manuscript, could MAI- PbI_2 -DMF be participating instead of PbI_2 -DMF? The authors state that NH_4Cl suppresses the PbI_2 -DMF phase. Is there any evidence for this?*

Response 5: We thank the referee for the insightful comment. We do not exclude the formation of MAI- PbI_2 -DMF phase during the precipitation process. We collected the precipitation from precursor solution (formed by injecting CB anti-solvent as shown in Figure 2h and Supplementary

Figure 9) and carried out XRD studies. We confirm the presence of $(MA)_2(DMF)_2Pb_mI_{2m+2}$ ($m=2,3$) together with PbI_2 -DMF as shown in the Supplementary Figure 4c. So we agree that the precipitate can be a mixture of PbI_2 -DMF and MAI- PbI_2 -DMF phases. This is reasonable because the $(MA)_2(DMF)_2Pb_mI_{2m+2}$ ($m=2,3$) is the intermediate phase for PbI_2 -DMF phase converted to bulk perovskites (Petrov, A. A. et al. J. Phys. Chem. C, 121, 20739-20743 (2017); Jung, M. et al. Chem. Soc. Rev., 48, 2011 (2019)). On the other hand, in the perovskite film spin coated from precursor solution without NH_4Cl as additives, i.e. the PbI_2 -based precipitation will become severe, excessive PbI_2 -DMF and $(MA)_2(DMF)_2Pb_3I_8$ was found remaining in the final film, which lead to much increased XRD peak intensity around $2\theta=9.63^\circ$ and $2\theta=6.63^\circ$, respectively (Supplementary Figure 10). In the revised manuscript, we define “ PbI_2 -DMF-contained solvated phase” as “PDS”, which involves PbI_2 -DMF and MAI- PbI_2 -DMF phases.

Supplementary Figure 10 | XRD patterns of unheated RP perovskite film with and without NH_4Cl additive, where the PbI_2 -DMF phase and $(MA)_2(DMF)_2Pb_3I_8$ become significant in the (unheated) samples w/o NH_4Cl additive.

The formation of MAI- PbI_2 -DMF phase didn't change our conclusion on the templated growth of 2D perovskite. MAI- PbI_2 -DMF phase like $(MA)_2(DMF)_2Pb_2I_6$ and $(MA)_2(DMF)_2Pb_3I_8$ are also needle-like crystal (Petrov, A. A. et al. J. Phys. Chem. C, 121, 20739-20743 (2017)). As shown in Figure Supplementary Figure 8, we intentionally soak $(MA)_2(DMF)_2Pb_3I_8$ fibers into the oversaturated BA based RP perovskite ($<n>=2$) solution (similar to the experiment shown in Figure 2 of main text), same directional growth of layered perovskite from the $(MA)_2(DMF)_2Pb_3I_8$ fibers can be observed.

This is because MAI-PbI₂-DMF phase also lead to the formation of 3D-like perovskite on its surface (black colored in Supplementary Figure 8) which is the key issue to trigger directional growth of layered perovskites.

For a better accuracy, we follow referee's suggestion and modified our description as follows:

In page 6, paragraph 2 of our revised manuscript, we modified the sentence of "by forming one-dimensional PbI₂-DMF solvate phase (PDS)" to be "by forming one-dimensional PbI₂-DMF-contained solvate phases (PDS)"

In page 6, paragraph 2 of our revised manuscript, we add a sentence of "The PDS formed in MAI-rich solution can be a mixture of PbI₂-DMF and (MA)₂(DMF)₂Pb_mI_{2m+2} (m=2,3) phases (see Supplementary Figure 4 and Supplementary Note 2), the latter of which has been reported to be the intermediate phase for the formation of perovskites"

In page 8, paragraph 1 of our revised manuscript, we add a sentence of "Replacing the PbI₂-DMF phase in Fig. 2d,g with (MA)₂(DMF)₂Pb₃I₈ phase or PbI₂-DMSO based solvated phase can also lead to the same directional growth of RP perovskites (Supplementary Figure 7 and Supplementary Figure 8), this is because those solvated phase has similar double/triple chains of edge-sharing PbI₆ octahedra to form 3D-like perovskite on its surface."

In page 29 of our Supplementary Information, we add a paragraph of "To determine the composition of the precipitation shown in Figure 3g, we collected the precipitation and carried out XRD measurement. The diffraction peaks of 2θ=9.06° and 2θ=9.63° proves the presence of PbI₂-DMF phase. Besides, intermediate phase of (MA)₂(DMF)₂Pb₂I₆ and (MA)₂(DMF)₂Pb₃I₈ were also observed, as indicated by the diffraction peaks marked in Supplementary Figure 4c. These (MA)₂(DMF)₂Pb_mI_{2m+2} (m=2,3) phases have been reported to be intermediate steps for the formation of bulk perovskites. As shown in Supplementary Figure 4d, solvated phase like (MA)₂(DMF)₂Pb₂I₆ and (MA)₂(DMF)₂Pb₃I₈ also possess one dimensional structures and needle-like crystals. "

In our Supplementary Information, we add the Supplementary Figure 4 and Supplementary Figure 8 as follows:

Supplementary Figure 4 | $\text{PbI}_2\text{-DMF}$ -contained solvated phases (PDS). (a) XRD spectrum of pure $\text{PbI}_2\text{-DMF}$ fibers. The predicted position of $(0\ 1\ \bar{1})$, $(0\ 2\ 0)$ and $(0\ \bar{2}\ 1)$ peaks of $\text{PbI}_2\text{-DMF}$ also shown (black lines) and inset showed photo of $\text{PbI}_2\text{-DMF}$ formed in $\text{PbI}_2\text{:DMF}$ solution at RT. (b) FTIR spectroscopy of $\text{PbI}_2\text{-DMF}$ crystal, indicating the presence of DMF molecules. (c) XRD patterns of the precipitation obtained from $\text{BA } \langle n \rangle = 4$ $0.5\text{NH}_4\text{Cl}$ precursor solution by antisolvent method, where the precipitation of PDS include $\text{PbI}_2\text{-DMF}$ and MAI- $\text{PbI}_2\text{-DMF}$ (i.e. $(\text{MA})_2(\text{DMF})_2\text{Pb}_2\text{I}_6$ and $(\text{MA})_2(\text{DMF})_2\text{Pb}_3\text{I}_8$.) phases. (d) Scheme of crystal structures of $\text{PbI}_2\text{-DMF}$, $(\text{MA})_2(\text{DMF})_2\text{Pb}_2\text{I}_6$ and $(\text{MA})_2(\text{DMF})_2\text{Pb}_3\text{I}_8$ solvated phases.

Supplementary Figure 8 | $(\text{MA})_2(\text{DMF})_2\text{Pb}_3\text{I}_8$ intermediate phase triggered templated growth of 2D perovskite. (a) XRD patterns of $(\text{MA})_2(\text{DMF})_2\text{Pb}_3\text{I}_8$ fibers with its photograph shown as inset. (b) Templated growth of 2D perovskite triggered by $(\text{MA})_2(\text{DMF})_2\text{Pb}_3\text{I}_8$ powders dropped on the surface of oversaturated 2D perovskite precursor solution. (c) Absorption and (d) Photoluminescence spectra of $(\text{MA})_2(\text{DMF})_2\text{Pb}_3\text{I}_8$ fibers after soaked in the oversaturated 2D perovskite precursor solution. (e) Inserting $(\text{MA})_2(\text{DMF})_2\text{Pb}_3\text{I}_8$ fibers into the oversaturated precursor solution to induce the templated growth of 2D perovskite. (f) Measured crystal size against growing time and the corresponding estimated growth speed of RP perovskites.

As for the suppressed nucleation of PbI_2 -DMF-contained solvated phases, in Figure 3c of the main text, we dissolved pristine PbI_2 into DMF solvent and increase the concentration gradually to see the critical concentration of precursor solution for the nucleation of PDS. We found the critical concentration for the nucleation of PbI_2 -DMF roughly doubled when NH_4Cl was used. Moreover, in Figure 3f,g of the main text, the adding of NH_4Cl and other additives (in perovskite precursor solution) suppressed the formation of precipitation, which are the PDS including PbI_2 -DMF and $(\text{MA})_2(\text{DMF})_2\text{Pb}_m\text{I}_{2m+2}$ ($m=2,3$) phases.

Figure 3c | Influence of the AX additives (AX/PbI₂=0.5) on the critical concentration of precursor solution for the nucleation of PbI₂-DMF solvated phase in DMF;

Comment 6: *The authors state that lattice-matching defines orientation. The authors do not have the evidence to make such strong claims.*

Response 6: We thank the referee for the comment. Generally, observing the quasi-3D perovskite/layered perovskite interface with high resolution transmission electron microscope (HRTEM) would be a straightforward method to demonstrate the lattice matching assumption. However, unfortunately, the facility is not accessible for us at this moment. So we conducted the templated growth of 2D perovskite (T-RP) on 3D MAPbI₃ single crystal (3D-SC) to observe how the facets of 3D-SC (i.e. with different lattice constant) impact on the orientation of macro RP perovskite crystals and verify the lattice matching:

Similar to the templated growth experiment shown in Figure 2d,g, templated growth of BA-based RP perovskite crystals on different facets of 3D-SC has been investigated by replacing PDS fibers with MAPbI₃ 3D-SC. [Figure citation redacted] If ignore the distortion of [PbI₆]⁴⁻ octahedral, the facets can be divided into two categories (Schlipf, J. et al., Sci. Rep-Uk 8, 4906 (2018)): one kind of facet is rectangle with the corners of [PbI₆]⁴⁻ octahedral pointing out, such as (1 1 0) or (0 0 2) facet; shown in [Figure citation redacted] the others are hexagon or pentagon with the edges of PbI₆ octahedral pointing out, such as (1 1 2), (0 2 0) or (2 0 0) facets shown in [Figure citation redacted]. Due to the different orientation of [PbI₆]⁴⁻ octahedral, these two kinds of facets have different out-of-plane diffraction peaks as shown in [Figure citation redacted]. Soaking 3D-SC in oversaturated RP precursor solution (n=2) for few minutes led to the templated growth of RP perovskites crystal (T-RP) arrays on every facet [Figure citation redacted], converting the surface of the 3D-SC from black color to red color.

[Figure redacted]

[redacted] Accordingly, we found the T-RP also shows two typical kinds of orientation:

i), On the facets with (1 1 2), (2 0 0) or (0 2 0) planes are exposed (i.e. the edges of PbI_6 octahedral pointing out), the T-RP sheets stand on the surface with a titling angle of $\sim 45^\circ$ to the terminal planes of 3D-SC (either be left or right tilted). [redacted] ii), On the facets with (1 1 0) or (0 0 2) planes are exposed (i.e. the corners of PbI_6 octahedral pointing out), the T-RP sheets stand vertically on the surface. [redacted]

[Figure redacted]

[redacted]

[Figure redacted]

As a conclusion, the templated growth of RP perovskite on 3D-SC experiment clearly demonstrated that the orientation of the $[\text{PbI}_6]^{4-}$ octahedral on 3D perovskite surface greatly determined the orientation of the $[\text{PbI}_6]^{4-}$ octahedral on RP perovskite, the lattice matching between 3D perovskite and layered perovskite play the key role in the final orientation of RP perovskites. We hope these additional studies can convince the referee.

[redacted]

Comment 7. *The authors state that the lowest energy configuration between the PbI_2 -phase and liquid is along the long side. Again, what evidence is there for this?*

Response 7: Thanks for the comment. To confirm the lowest energy configuration, we intentionally dropped some needle-like PbI_2 -DMF crystals in micrometer scale on the top of the saturated RP perovskite precursor solution for the optical observation. As shown in Figure R8, the needle-like PbI_2 -DMF crystals lying on the liquid top with its length direction parallel to liquid/air interface. This is reasonable if taking the surface tension into consideration, because generally liquid tends to minimize its surface area due to its surface tension, while the lying of PbI_2 -DMF crystals on top of liquid helps to minimize the surface area of the liquid phase than other possible configurations as illustrated in Figure R8c.

Figure R8 | Alignment of micrometer sized needle-like PDS crystal on the top of the saturated RP perovskite precursor solution. (a) Illustration of the introduction of needle-like PbI_2 -DMF crystal onto the top of saturated RP perovskite precursor solution; (b) Photography of the lying of the needle-like PbI_2 -DMF crystals at the liquid/air interface observed under optical microscope; (c) A proposed model for the explanation of the lowest energy configuration between the PbI_2 -DMF and liquid, in which the lying of PbI_2 -DMF crystals on liquid surface is energetically favored because it can lead to reduced surface area and hence surface energy of the liquid phase.

Comment 8. Overall, the writing can be substantially improved. There are a number of instances of clerical errors, poor word choice, and informal language (e.g., “a bunch of”, “it turns out”, “warranties”).

Response 8: Thanks for the comments.

In page 5, paragraph 1 of our revised manuscript, we changed to sentence of “It turns out that the rough TiO_2 surface ...” to be “It can be confirmed that the rough TiO_2 surface ...”

In page 7, line 16, we replaced “PDS fiber made of a bunch of PDS nanowires” as “PDS fiber made of PDS nanowires”;

In page 11, paragraph 1 of our revised manuscript, we changed the sentence of “It turns out the lattice matching between the facets of solvated perovskites and layered perovskite defines the crystal orientation” to be “It can be noticed that the lattice matching between the facets of quasi-3D perovskite and layered perovskite may play the key role in defining the crystal orientation”;

In page 8, paragraph 1 of our revised manuscript, we changed the sentence of “The high speed warranties the formation of RP perovskites within one second” to be “The high speed enables the formation of RP perovskites within one second”.

The language in our revised manuscript was double checked now.

Reviewer #2:

Comment 1: *In this manuscript, Wang et al. report the templated growth of layered perovskites with out-of-plane orientation on solvated perovskites. The growth mechanism demonstrated in this work is reasonably proved by experimental data, which provides insightful understanding on the growth of vertically oriented layered perovskite films using perovskite precursor solutions with additional ammonium halide salts. In addition, the presented RP solar cells show device performance comparable to the best of the reported layered solar cells. The manuscript is well written and would be of great interest to the perovskite community. Therefore, I strongly recommend the publication of this work in Nature Communications. The following comments are provided to the authors to further strengthen the manuscript.*

When soaked in oversaturated RP precursor solution, the surface of PDS powders turn black in a few seconds (Supplementary Figure 5). Then the surface of the PDS induces the growth of oriented RP perovskites. However, it is unclear when the PDS totally turns into perovskite phase. Does it occur during solution thinning, after spin-coating or during thermal annealing?

Response 1: We appreciate the referee's very positive comments and suggestions, which helped us to improve the quality of this manuscript tremendously.

The "PDS" in our previous manuscript represent PbI_2 -DMF phase. In the revised manuscript, the $(\text{MA})_2(\text{DMF})_2\text{Pb}_m\text{I}_{2m+2}$ ($m=2,3$) phases are also involved in our analysis. So here the PDS include the PbI_2 -DMF phase and $(\text{MA})_2(\text{DMF})_2\text{Pb}_m\text{I}_{2m+2}$ ($m=2,3$) phases in our revised manuscript. It's still difficult for us to carry out in-situ study on the conversion of PDS to perovskite phase. We found the PbI_2 -DMF phase is stable at RT (e.g. see Figure 2 of our manuscript), and the $(\text{MA})_2(\text{DMF})_2\text{Pb}_m\text{I}_{2m+2}$ ($m=2,3$) phases are also reported stable at RT (Petrov, A. A. et al. J. Phys. Chem. C, 121, 20739-20743 (2017)). So it's not necessarily all the PDS will be converted into perovskite during solution thinning, depending on the dynamic of DMF evaporation. For the precursor solution with NH_4Cl additives, the amount of precipitated PDS before the growth of RP perovskite is largely reduced, and most of the PDS phase should be converted to perovskite during solution thinning. Due to this reason, we do not found the presence of PbI_2 -DMF phase or $(\text{MA})_2(\text{DMF})_2\text{Pb}_m\text{I}_{2m+2}$ ($m=2,3$) phases by XRD measurement after spin coating process (Supplementary Figure 10a). Nevertheless, we could like to mention that remaining of PbI_2 -DMF or MAI- PbI_2 -DMF related phase is thermodynamically possible. For the precursor solution without NH_4Cl additives, much more PDS phases would be formed before the growth of RP perovskites (see the illustration in Figure 3a). In this case, there is a possibility that some of the PDS phases are not converted to perovskite during solution thinning. As shown in Supplementary Figure 10b, the peak around $2\theta=9.5^\circ$ indicates the presence of residual PbI_2 -DMF phase in sample spun from precursor without NH_4Cl additives, and the XRD diffraction peaks around $2\theta=6.6^\circ$ and $2\theta=8.1^\circ$ also indicate the presence of $(\text{MA})_2(\text{DMF})_2\text{Pb}_3\text{I}_8$ as the intermediate phase for the formation of bulk perovskites (see Ref. Petrov, A. A. et al. J. Phys. Chem. C 121, 20739-20743 (2017)). These remained PbI_2 -DMF and

(MA)₂(DMF)₂Pb₃I₈ phases will totally converted to perovskite phase after a thermal annealing at 100 °C, so that no PbI₂-DMF and (MA)₂(DMF)₂Pb₃I₈ phases can be detected in the final RP perovskite films as shown in Supplementary Figure 11a and Figure 4f in our revised manuscript.

In page 9, paragraph 2 of our revised manuscript, we add the following sentence “It’s not necessarily all the PDS will be converted into perovskite phase during solution thinning, depending on the dynamic of DMF evaporation. So that the PbI₂-DMF and (MA)₂(DMF)₂Pb_mI_{2m+2} phases can be detected in spin coated film (Supplementary Figure 10) until heated at elevated temperature of 70~100 °C.”

In page 10, paragraph 1 of our revised manuscript, we add the following sentence “Since the preformed PDS are much suppressed by NH₄Cl additives, these residual solvated phase may not be necessarily detectable by XRD (Supplementary Figure 10).”

In our Supplementary Information, we add the Supplementary Figure 10 as follows:

Supplementary Figure 10 | XRD patterns of unheated RP perovskite films with and without NH₄Cl additive, where the PbI₂-DMF and (MA)₂(DMF)₂Pb₃I₈ phases become significant in the samples w/o NH₄Cl additive.

Comment 2: *The PDS (PbI₂-DMF solvated phase) plays a critical role in the growth of oriented layered perovskites. Could the DMF in PDS be other solvent molecules, e.g. DMSO?*

Response 2: According to referee’s suggestion, we further used PbI₂-DMSO based solvated phase

instead of PbI_2 -DMF phase to reproduce the experiment shown in Figure 2 of our manuscript. As shown in Supplementary Figure 7 of our revised manuscript, 3D-like perovskite with black color can also be formed on PbI_2 -DMSO based solvated phase immediately when it is soaked in an oversaturated precursor solution, and then similar directional growth of RP perovskite triggered by quasi-3D perovskite can be observed. The 2D perovskite grows with an estimated speed of $0.72 \mu\text{m/s}$, which is in the same range of the growth speed in the case based on PbI_2 -DMF phase (Supplementary Figure 6 of our revised manuscript) or $(\text{MA})_2(\text{DMF})_2\text{Pb}_3\text{I}_8$ phase (Supplementary Figure 8f). So we conclude that the templated growth can be triggered by PbI_2 -DMSO based solvated phase as well.

In page 8, paragraph 1 of our revised manuscript, we add a sentence of “Replacing the PbI_2 -DMF phase in Fig. 2d,g with $(\text{MA})_2(\text{DMF})_2\text{Pb}_3\text{I}_8$ phase or PbI_2 -DMSO based solvated phase can also lead to the same directional growth of RP perovskites (Supplementary Figure 7 and Supplementary Figure 8), this is because those solvated phases have similar double/triple chains of edge-sharing PbI_6 octahedra to form 3D-like perovskite on its surface.”

Supplementary Figure 7 | Templated growth of 2D perovskite on PbI_2 -DMSO based solvated phase. Illustration and photos of the templated growth of BA-based RP crystals on (a) A PbI_2 -DMSO solvated phase fiber soaked in the oversaturated precursor solution ($\langle n \rangle = 2$) and (b) PbI_2 -DMSO solvated phase powders dropped on the surface of the oversaturated precursor solution. (c) Absorption and (d) photoluminescence spectra of PbI_2 -DMSO solvated phase fiber before and after being soaked in oversaturated 2D perovskite. (e) Estimated growth rate ($\sim 0.72 \mu\text{m s}^{-1}$) of 2D perovskites on PbI_2 -DMSO solvated phase fiber.

Supplementary Figure 8 | $(\text{MA})_2(\text{DMF})_2\text{Pb}_3\text{I}_8$ intermediate phase triggered templated growth of 2D perovskite. (a) XRD patterns of $(\text{MA})_2(\text{DMF})_2\text{Pb}_3\text{I}_8$ fibers with its photograph shown as inset. (b) Templated growth of 2D perovskite triggered by $(\text{MA})_2(\text{DMF})_2\text{Pb}_3\text{I}_8$ powders dropped on the surface of oversaturated 2D perovskite precursor solution. (c) Absorption and (d) Photoluminescence spectra of $(\text{MA})_2(\text{DMF})_2\text{Pb}_3\text{I}_8$ fibers after soaked in the oversaturated 2D perovskite precursor solution. (e) Inserting $(\text{MA})_2(\text{DMF})_2\text{Pb}_3\text{I}_8$ fibers into the oversaturated precursor solution to induce the templated growth of 2D perovskite. (f) Measured crystal size against growing time and the corresponding estimated growth speed of RP perovskites.

Comment 3: Ammonium halide salts such as NH_4Cl improve the solubility of PDS, which has been proved by the authors. Could the authors further explain why these salts can improve the solubility of PDS?

Response 3: Thanks for the recognition. We are not quite sure why the solubility of PbI_2 can be improved by excessive AX salts, here we try to propose a possible reason for consideration: perovskite precursor solutions for solar cells were proved as generally colloidal dispersions in

precursor solution (Yan, K. et al., J. Am. Chem. Soc. 137, 4460-4468 (2015)). According to P. Kamat group's results (K. Stamplecoskie et al., Energy Environ. Sci., 8, 208 (2015)), the Pb^{2+} ions in precursor solution can form plumbate complexes with halide ions (X^-) such as PbI_3^- and PbI_4^{2-} . With the presence of excessive AX salt, there are increasing in the concentration of halide ions, which would lead to higher degree halide complex with more negative charges on the Pb-I based colloidal particles. When the Pb-I based colloidal particles become more negatively charge, due to the electrostatic repelling force, the aggregation of the Pb-I based colloidal particles in solution become difficult. The stabilization of Pb-I based colloidal particles in DMF by excessive AX salts can be one of the possible reason for the increased solubility (Tong, G. et al. Materials Today Energy 5, 173-180 (2017)).

Comment 4: *As reported in literature, e.g. Adv. Energy Mater. 2018, 8 (21), 1800185 and J. Am. Chem. Soc. 2017, 139, 1432-1435, RP perovskite films prepared from nominal n values are actually mixtures with various RP phases. These RP perovskite films show graded phase distribution even with out-of-plane orientation. Supplementary Figure 17 in this work also shows a similar observation. Could the authors comment on the growth mechanism of graded RP phases with out-of-plane orientation?*

Responses 4: Thanks for raising this issue which are also very general phenomenon. We believe this frequently observed vertical phase separation can be understood with our results. In our explanation, the downward growth of the RP perovskite was confirmed as the major crystal growth mechanism. So the vertical phase separation is closely related to the nonuniform materials supply (i.e. the precipitated raw materials from dissolved state) during solution thinning. We confirmed the solubility of BAI, MAI and PbI_2 in DMF follow an order of $\text{PbI}_2 < \text{MAI} < \text{BAI}$, as shown in Supplementary Figure 2b. This lead to PbI_2 -DMF and MAI- PbI_2 -DMF solvated phase preferably formed on top of the resulted film, which turns to perovskite with large n value or 3D perovskite phases in the lateral thermal annealing process. And the much higher solubility of BAI make the layered perovskite with low n value preferably formed on bottom of the resulted film. Although thermal annealing can promote the interdiffusion of these species, a detectable graded vertical phase separation is still expected.

In page 15, paragraph one of our revised manuscript, we changed the last sentence "At the last stage of spin coating or doctor blading, the solvated phase on top converted to a thin layer consisted of larger-n RP perovskites due to the excessive MA⁺ cations and lacked BA⁺ cation in the preformed solvated phase, contributing to the frequently observed vertical phase separation with large-n RP perovskites on top" to be "Due to the solubility difference shown in Figure 2b, there should be a preferably precipitation of PbI_2 -DMF and MAI- PbI_2 -DMF solvated phase on top and BA-rich phase on bottom during spin coating or doctor blading, which leads to relative more larger-n RP perovskites formed on top of the resulted film, contributing to the frequently observed vertical phase separation"; and we added the reference of "J. Am. Chem. Soc., 139, 1432-1435 (2017)" and "Adv. Energy Mater., 8, 1800185 (2018)" as Ref. 38 and Ref.

39 in our revised manuscript.

Fig. 2b | Comparison of the solubility of Pbl₂, MAI and BAI in DMF solvent

Reviewer #3:

Comment 1: The manuscript by Wang et al. presents a thorough investigation of the growth mechanism for oriented layered perovskites. The main finding is that the firstly formed solvated perovskites in solution can template the sequential growth of layered perovskites, the orientation of layered perovskites is determined by the lattice matching between solvated perovskite and layered perovskites. In my opinion, since H. Tsai et al. reported hot-cast RP-type layered perovskites with out-of-plane orientation (Nature 536, 312, (2016)), the detailed nucleation process of layered perovskites is still unclear. The finding in this manuscript is very exciting to the 2D perovskite field, because the authors clarified several unsolved questions with this templated growth mechanism, especially how the out-of-plane orientation was formed. Moreover, it was well demonstrated that the formation of solvated perovskites can be engineered, which could be a useful method to achieve layered perovskite films with out-of-plane orientation and hence improve device performance. The manuscript is well written and organized with ample relevant references. I consider this work to be suitable for the broad readership in Nature Communications and I would recommend publication of this manuscript after revision.

The following are some specific concerns: According to some previous results (e.g. K. Yan et al. J. Am. Chem. Soc. 137, 4460 (2015)), the perovskite precursor solutions are generally colloidal dispersions, so I am wondering whether the excessive NH_4Cl also impacted on the colloidal size and hence tuned the grain size and final morphology of the layered perovskite films?

Response 1: We appreciate the referee's recognition and very positive comments.

Figure R9 | Particle size distribution of RP perovskite precursor solution with and without NH_4Cl , where adding NH_4Cl donot impact a lot on the particle size of colloidal in our perovskite solution.

We check the colloidal size in our precursor solution with and without additive by dynamic laser scattering method with 780 nm laser as light source (Nanotracc Wave II, Microtrac), as shown in Figure R9. We find that the particle size of colloidal in our perovskite solution with and without additive are all less than 3 nm, which is different with the cased reported in by Tong, G. et al.

(Materials Today Energy 5, 173-180 (2017)). The absence of colloid particles with large diameter might be due to the different purity of PbI_2 raw materials (99.999%) and supplier (Sigma-Aldrich) in our case. Since the distributions of the particle size are very close for both cases, we can confirm that the increased crystallinity and grain orientation by the NH_4Cl additives is not directly related to the changes in colloidal sizes.

Comment 2: *It is impressive that the out-of-plane orientation and high crystallinity of the RP perovskites can be equally achieved in a group of films with excessive AX salts ($A = \text{NH}_4^+$ or MA^+ ; $X = \text{Cl}^-$, Br^- or I^-) as additives. So how about the device performance for solar cells resulted from precursor solutions with other AX salts than NH_4Cl ?*

Response 2: We thank referee for raising this issue. Although many other AX salts can lead to similar OP orientation of RP perovskites, in our study, we decided to choose NH_4Cl for the systematic device optimizing. This is because NH_4Cl additives delivered the best PCE, as show in Figure R10. We note that both NH_4^+ ions and Cl^- ions are too small to be stable in perovskite phase, so that this kind of additive is readily to be removed during thermal annealing, which has been confirmed by our XPS analysis (Figure R2) and agree with some published works (Si, H. N. et al. Adv. Funct. Mater., 27, 1701804 (2017); Xie, F. X. et al. ACS Nano, 9, 639-646 (2015)). The easy evaporation of NH_4Cl helps to minimize the influence of additives on the stoichiometric ratio of RP perovskites and hence its average n-values, which is superior than using excessive MA^+ ions, or excessive Br^- or I^- ions.

Figure R10 | Performance of RP perovskite solar cells with different additives. J - V characteristics recorded from the BA $\langle n \rangle = 4$ layered perovskite with NH_4Cl , MACl , NH_4I , and MAI as additives respectively.

Figure R2 | X-ray photoelectron spectroscopy (XPS) of BA based layered perovskite ($\langle n \rangle = 4$) fabricated from precursor solution with 0.5 molar ratio of NH_4Cl additive. The absence of Cl 2p core level peak around 200 eV suggests undetectable Cl^- ions remained in the perovskite film.

Comment 3: In Fig. S14a, the XRD pattern of RP films with different amounts of NH_4Cl addition was shown, the absence of diffraction peaks below 10° when the molar ratio of $\text{NH}_4\text{Cl}:\text{PbI}_2 > 0.3$ was believed to be due to the dominating OP orientation induced by NH_4Cl additive. However, there is an exception for 1.0 NH_4Cl , in which some extra diffraction peaks show up. Could the authors provide some discussion on this?

Response 3: Thanks for pointing this phenomenon out. We agree that the peaks below 10° in Supplementary Figure 17a (i.e. the Fig. S14a in our previous version) indicating IP orientation become detectable when too much NH_4Cl was used (e.g. $\text{NH}_4\text{Cl}:\text{PbI}_2 = 1.0$). From the cross-section SEM image shown in Figure R11, crystal grain with random orientation can be observed when too much NH_4Cl additive was used. Nevertheless, we can still explain it in the frame of solvated phased triggered templated growth of RP perovskites. In our story, the nucleation of RP perovskite is strongly determined by the location and amounts of preformed PDS phase, which dominated the finally crystal orientation and morphologies. However, the amount of PDS precipitation can be much reduced by NH_4Cl additives. There should be an optimized range for the loading of NH_4Cl , i.e. if there are too much NH_4Cl additive in precursor solution, the preformed PDS precipitation will be insufficient to cover the whole liquid-air interface, and then the templated growth of RP perovskite will become nonuniform, or alternatively, become noncompetitive with homogenous nucleation inside the liquid phase. As a consequence, grains with IP or random orientation will show up. We

believe this unfavorable IP orientation can be avoided by carefully select an optimized loading of NH_4Cl additive.

Figure R11. Cross-section SEM image of RP layered perovskite based on BA $\langle n \rangle = 4$ with $\text{NH}_4\text{Cl}/\text{PbI}_2 = 1.0$.

In the legend of Supplementary Figure 17 of our revised SI, we add the sentence of “Some IP orientation of RP perovskite can be found at the point of $\text{NH}_4\text{Cl}/\text{PbI}_2 = 1.0$, which might be due to the excessive NH_4Cl additive severely suppressed the preformed PDS, making the templated growth of RP perovskite become nonuniform, or alternatively, noncompetitive with homogenous nucleation inside the liquid phase.”

Reviewers' Comments:

Reviewer #1:

Remarks to the Author:

In this revised manuscript, the authors have addressed the major issues using additional experiments and analyses:

1. They now provide a set of optical studies that prove that the absence of low-angle diffraction peaks along qz direction in the GIWAXS is a result of out-of-plane oriented growth of layered perovskites.
2. They further clarify – through the XRD study of precipitates within precursors – the real phase of nucleation sites which promote the oriented growth.
3. A new set of single crystal studies is also added and it further supports the claim that the lattice matching between layered perovskite and 3D perovskite is the main driving force for templated growth. Different facets of 3D perovskites induce different growth direction of layered perovskites.

With these revisions, I agree with the authors' conclusion that minimizing the crystal nucleation within precursors induces out-of-plane oriented growth of layered perovskites. I recommend this manuscript for publication in Nature Communications.

Reviewer #2:

Remarks to the Author:

The authors have properly solved my concerns and the quality of the revised manuscript is improved. It is suitable for publication.

Reviewer #3:

Remarks to the Author:

The author has addressed all my concerns and the quality of the manuscript has been greatly improved. Hence, I recommend to publish the paper in Nature Communications without further revision.